# Local Learning for Covariate Selection in Nonparametric Causal Effect Estimation with Latent Variables

**Zheng Li**[1,2], **Xichen Guo**[1], **Feng Xie**[1,*], **Yan Zeng**[1], **Hao Zhang**[2,*], **Zhi Geng**[1]

[1]Department of Applied Statistics, Beijing Technology and Business University, Beijing, China
[2]SIAT, Chinese Academy of Sciences, Shenzhen, China

## Abstract

Estimating causal effects from nonexperimental data is a fundamental problem in many fields of science. A key component of this task is selecting an appropriate set of covariates for confounding adjustment to avoid bias. Most existing methods for covariate selection often assume the absence of latent variables and rely on learning the global causal structure among variables. However, identifying the global structure can be unnecessary and inefficient, especially when our primary interest lies in estimating the effect of a treatment variable on an outcome variable. To address this limitation, we propose a novel local learning approach for covariate selection in nonparametric causal effect estimation, which accounts for the presence of latent variables. Our approach leverages testable independence and dependence relationships among observed variables to identify a valid adjustment set for a target causal relationship, ensuring both soundness and completeness under standard assumptions. We validate the effectiveness of our algorithm through extensive experiments on both synthetic and real-world data.

## 1 Introduction

Estimating causal effects is crucial in various fields such as epidemiology [Hernán and Robins, 2006], social sciences [Spirtes et al., 2000], economics [Imbens and Rubin, 2015], and artificial intelligence [Peters et al., 2017, Chu et al., 2021]. In these domains, understanding and accurately estimating causal relationships are vital for policy-making, clinical decisions, and scientific research. Within the framework of causal graphical models, covariate adjustment, such as the use of the back-door criterion [Pearl, 1993], emerges as a powerful and primary tool for estimating causal effects from observational data, since implementing idealized experiments in practice is difficult [Pearl, 2009]. Formally speaking, let $do(x)$ stand for an idealized experiment or intervention, where the values of $X$ are set to $x$, and $f(y|do(x))$ denote the causal effect of $X$ on $Y$. A valid covariate is a set of variables $\mathbf{Z}$ such that $f(y \mid do(x)) = \int_{\mathbf{z}} f(y \mid x, \mathbf{z})f(\mathbf{z})d\mathbf{z}$ [Pearl, 2009, Shpitser et al., 2010]. Consider the graph (a) in Figure. 1, $\mathbf{Z} = \{V_5\}$ is a valid covariate set w.r.t. (with respect to) the causal relationship $X \to Y$.

Given a causal graph, one can determine whether a set is a valid adjustment set using adjustment criteria such as the back-door criterion [Pearl, 1993, 2009]. The main challenge in covariate adjustment estimation is to find a valid covariate set that satisfies the back-door criterion using only observational data, without prior knowledge of the causal graph. To tackle this challenge, Maathuis et al. [2009] proposed the IDA (Intervention do-calculus when the DAG (Directed Acyclic Graph) is Absent) algorithm. This algorithm first learns a CPDAG (Complete Partial Directed Acyclic Graph)

---

*Corresponding authors: Feng Xie <fengxie@btbu.edu.cn>, Hao Zhang <h.zhang10@siat.ac.cn>

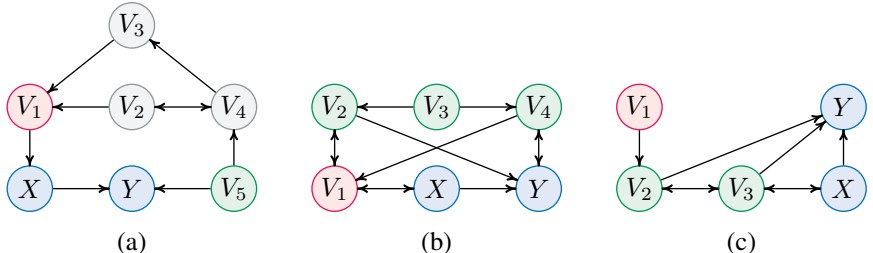

Figure 1: Example MAGs with treatment $X$ and outcome $Y$. Nodes shaded in green represent a valid adjustment set. (a) Both global search EHS and local search CEELS methods identify the adjustment set. (b) Adapted from Cheng et al. [2022], CEELS fails to select the adjustment set despite the presence of a COSO variable $V_1$ (See Fig. 4 in Cheng et al. [2022]). (c) An example without a COSO variable, where the adjustment set can still be found locally.

using the PC (Peter-Clark) algorithm [Spirtes and Glymour, 1991], enumerates all Markov equivalent DAGs, and estimates all possible causal effects of a treatment on an outcome. Additionally, with domain knowledge about specific causal directions, one can further identify more precise causal effects [Perkovic et al., 2017, Fang and He, 2020]. For instance, Perkovic et al. [2017] proposed the semi-local IDA algorithm, which provides a bound estimation of a causal effect when some directed edge orientation information is available. To efficiently find covariates, a local method CovSel utilizes criteria from De Luna et al. [2011] for covariate selection [Häggström et al., 2015]. Though these methods have been used in a range of fields, they may fail to produce convincing results in cases with latent confounders, as they do not properly take into account the influences from latent variables [Maathuis and Colombo, 2015].

There exists work in the literature that attempts to select covariates and estimate the causal effect in the presence of latent variables. Malinsky and Spirtes [2017] introduced the LV-IDA (Latent Variable IDA) algorithm based on the generalized back-door criterion [Maathuis and Colombo, 2015]. This algorithm initially learns a Partial Ancestral Graph (PAG) using the FCI (Fast Causal Inference) algorithm [Spirtes et al., 2000], then enumerates all Markov equivalent Maximal Ancestral Graphs (MAGs), and estimates all possible causal effects of a treatment on an outcome. Subsequently, Hyttinen et al. [2015] proposed the CE-SAT (Causal Effect Estimation based on SATisfiability solver) method, which avoids enumerating all MAGs in the PAG. Although these algorithms are effective, learning the global causal graph is often unnecessary and wasteful when we are only interested in estimating the causal effects of specific relationships.

Several contributions have been developed to select covariates for estimating causal effects of interest without learning global causal structure. For instance, Entner et al. [2013] designed two inference rules and proposed the EHS algorithm (named after the authors' names) to determine whether a treatment has a causal effect on an outcome. If a causal effect is present, these rules help identify an appropriate adjustment set for estimating the causal effect of interest, based on the conditional independencies and dependencies among the observed variables. The EHS method has been demonstrated to be both sound and complete for this task. However, it is computationally inefficient, with time complexity of $\mathcal{O}(t \times 2^t)$, where $t$ is the number of observed covariates. It requires an exhaustive search over all combinations of variables for the inference rules. More recently, by leveraging a special variable, the Cause Or Spouse of the treatment Only (COSO) variable, combined with a pattern mining strategy Agrawal et al. [1994], Cheng et al. [2022] proposed a local algorithm, called CEELS (Causal Effect Estimation by Local Search), to select the adjustment set. Although the CEELS method is faster than the EHS method, it may fail to identify an adjustment set during the local search that could be found using a global search. For instance, considering the causal graphs (b) and (c) illustrated in Figure 1, where $\{V_2, V_3, V_4\}$ and $\{V_2, V_3\}$ are the valid adjustment sets w.r.t. the causal relationship $X \rightarrow Y$, respectively. The CEELS algorithm fails to select these corresponding adjustment sets, whereas the EHS method is capable of identifying them.

It is desirable to develop a sound and complete local method to select an adjustment set for a causal relationship of interest. Specially, we make the following contributions:

1. We propose a novel, fully local algorithm for selecting covariates in nonparametric causal effect estimation, utilizing testable independence and dependence relationships among the observed variables, and allowing for the presence of latent variables.

2. We theoretically demonstrate that the proposed algorithm is both sound and complete, and can identify a valid adjustment set for a target causal relationship (if such a set exists) under standard assumptions, comparable to global methods.
3. We demonstrate the efficacy of our algorithm through experiments on both synthetic and real-world datasets.

## 2 Preliminaries

### 2.1 Definitions and Notations

**Graph.** A graph $\mathcal{G} = (\mathbf{V}, \mathbf{E})$ consists of a set of nodes $\mathbf{V} = \{V_1, \ldots, V_p\}$ and a set of edges $\mathbf{E}$. A graph $\mathcal{G}$ is **directed mixed** if the edges in the graph are **directed** ($\rightarrow$), or **bi-directed** ($\leftrightarrow$). A **causal path** (directed path) from $V_i$ to $V_j$ is a path composed of directed edges pointing towards $V_j$, i.e. , $V_i \rightarrow \ldots \rightarrow V_j$. A **non-causal path** from $V_i$ to $V_j$ is a path where at least one edge has an arrowhead at the mark near $V_i$. e.g. , $V_i \leftarrow V_{i+1} \leftarrow \ldots \rightarrow V_{j-1} \rightarrow V_j$. A path $\pi$ from $V_i$ to $V_j$ is a **collider path** if all the passing nodes are colliders on $\pi$, e.g. , $V_i \rightarrow V_{i+1} \leftrightarrow \ldots \leftrightarrow V_{j-1} \leftarrow V_j$. $V_i$ is called an **ancestor** of $V_j$ and $V_j$ is a **descendant** of $V_i$ if there is a causal path from $V_i$ to $V_j$ or $V_i = V_j$. A directed mixed graph is called an **ancestral graph** if the graph does not contain any directed or almost directed cycles. An ancestral graph is a **maximal ancestral graph** (*MAG*, denoted by $\mathcal{M}$) if there exists a set of nodes that m-separates any two non-adjacent nodes. A MAG is a *DAG* if it contains only directed edges. A directed edge $X \rightarrow Y$ in $\mathcal{M}$ is **visible** if there is a node $S$ not adjacent to $Y$, such that there is an edge between $S$ and $X$ that is into $X$, or there is a collider path between $S$ and $X$ that is into $X$ and every non-endpoint node on the path is a parent of $Y$. See Figure 8 in Appendix B for an example. Otherwise, $X \rightarrow Y$ is said to be **invisible**. A visible edge $X \rightarrow Y$ means that there are no latent confounders between $X$ and $Y$. All directed edges in DAGs are said to be visible. To save space, the detailed graph-related definitions are provided in Appendix B.

**Markov Blanket.** The *Markov blanket* of a variable $Y$ is the smallest set conditioned on which all other variables are probabilistically independent of $Y$ [1]. Graphically, in a MAG, the **Markov blanket** of a node $Y$, denoted $MB(Y)$, is unique and comprises: 1) the adjacent nodes of $Y$; and 2) all the non-adjacent nodes that have a collider path to $Y$ in the MAG. Figure 2 specifically illustrates the Markov blanket of the node $Y$ in the MAG. The nodes shaded in green belong to $MB(Y)$.

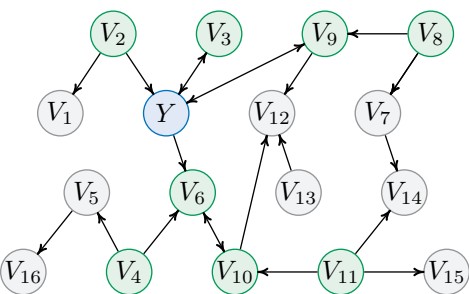

Figure 2: The illustrative example for MB in a MAG, where $Y$ is the target of interest and the green nodes belong to $MB(Y)$.

**Notations.** We use $Adj(V_i)$, $Pa(V_i)$, and $De(V_i)$, to denote the set of *adjacent, parents* and *descendants* of node $V_i$, respectively. We denote by $(X, Y)$ an ordered variable (node) pair, where $X$ is the **treatment** and $Y$ the **outcome**. We denote $\mathbf{X} \perp\!\!\!\perp \mathbf{Y}|\mathbf{Z}$ as "$\mathbf{X}$ is statistically independent of $\mathbf{Y}$ given $\mathbf{Z}$". Similarly, $\mathbf{X} \not\perp\!\!\!\perp \mathbf{Y}|\mathbf{Z}$ denotes that $\mathbf{X}$ is not statistically independent of $\mathbf{Y}$ given $\mathbf{Z}$. The main symbols used in this paper are summarized in Table 1 in the Appendix B.

### 2.2 Adjustment Set

The covariate adjustment method is often used to estimate causal effects from observational data [Pearl, 2009]. Throughout, we focus on the causal effect of a single treatment variable $X$ on the

---

[1] See Appendix B.2 for more details of Markov blanket.

single outcome variable $Y$. We next introduce a more general graphical language to describe the covariate adjustment criterion, namely the generalized adjustment criterion [Perkovi et al., 2018]. Before providing its definition, we first introduce two important concepts in the graph, as they will be used in the description of this definition.

**Definition 1** (Amenability [Van der Zander et al., 2014, Perkovi et al., 2018]). *Let $(X, Y)$ be an ordered node pair in a MAG. The MAG is said to be adjustment amenable w.r.t. $(X, Y)$ if all causal paths from $X$ to $Y$ start with a visible directed edge out of $X$.*

**Definition 2** (Forbidden set; $Forb(X, Y)$ [Perkovi et al., 2018]). *Let $(X, Y)$ be an ordered node pair in a DAG, or MAG $\mathcal{G}$. Then the forbidden set relative to $(X, Y)$ is defined as $Forb(X, Y) = \{W^{'} \in \mathbf{V} \mid W^{'} \in De(W), W$ lies on a causal path from $X$ to $Y$ in $\mathcal{G}\}$.*

**Definition 3** (Generalized adjustment criterion [Perkovi et al., 2018]). *Let $(X, Y)$ be an ordered node pair in a DAG or MAG $\mathcal{G}$. A set of nodes $\mathbf{Z} \subseteq \mathbf{V} \setminus \{X, Y\}$ satisfies the generalized adjustment criterion relative to $(X, Y)$ in $\mathcal{G}$ if $\mathcal{G}$ is adjustment amenable relative to $(X, Y)$, $\mathbf{Z} \cap Forb(X, Y) = \emptyset$, and all definite status non-causal paths from $X$ to $Y$ are blocked by $\mathbf{Z}$. If these conditions hold, then the causal effect of $X$ on $Y$ is identifiable and is given by [2]*

$$f(y \mid do(x)) = \begin{cases} f(y \mid x) & if\ \mathbf{Z} = \emptyset, \\ \int_{\mathbf{z}} f(y \mid x, \mathbf{z}) f(\mathbf{z}) d\mathbf{z} & otherwise. \end{cases} \tag{1}$$

Note that the generalized adjustment criterion is equivalent to the *generalized back-door criterion* of Maathuis and Colombo [2015] when the treatment $\mathbf{X}$ is a singleton[Perkovi et al., 2018]. Thus, condition 3 can be represented by the requirement that all definite status back-door paths from $X$ to $Y$ are blocked by $\mathbf{Z}$ in $\mathcal{G}$.

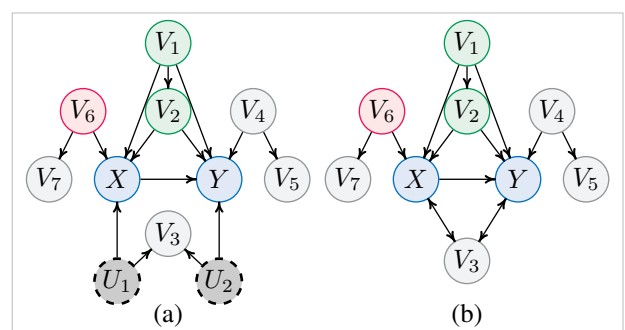

Figure 3: (a) An underlying causal DAG (adapted from Häggström [2018]), in which $U_1$ and $U_2$ are unobserved variables. (b) The corresponding MAG of the DAG shown in (a).

**Example 1** (Generalized adjustment criterion). *Consider the causal diagram shown in Figure. 3 (b). According to Definition 2, the MAG satisfies the amenability condition relative to $(X, Y)$, and $Forb(X, Y) = \{Y\}$ holds true in the graph. Then, the set $\{V_1, V_2\}$ is a valid adjustment set since, they can all block non-causal paths from $X$ to $Y$.*

## 2.3 Problem Definition

We consider a Structural Causal Model (SCM) as described by Pearl [2009]. The set of variables is denoted as $\mathbf{V} = \{X, Y\} \cup \mathbf{O} \cup \mathbf{U}$, with a joint distribution $P(\mathbf{V})$. Here, $\mathbf{O}$ represents the set of observed covariates, and $\mathbf{U}$ denotes the set of latent covariates. Therefore, the SCM is associated with a DAG, where each node corresponds to a variable in $\mathbf{V}$, and each edge represents a function $f$. Specifically, each variable $V_i \in \mathbf{V}$ is generated as $V_i = f_i(Pa(V_i), \varepsilon_i)$, where $Pa(V_i)$ denotes the parents of $V_i$ in the DAG, and $\varepsilon_i$ represents errors (or "disturbances") due to omitted factors. In addition, all errors are assumed to be independent of each other. Analogous to Entner et al. [2013], Cheng et al. [2022], we assume that $Y$ is not a causal ancestor of $X$ [3], and that $\mathbf{O}$ is a set of pretreatment variables w.r.t. $(X, Y)$, i.e., $X$ and $Y$ are not causal ancestors of any variables in $\mathbf{O}$.

**Remark 1.** *It is noteworthy that existing methods commonly employ the pretreatment assumption [Cheng et al., 2022, Entner et al., 2013, De Luna et al., 2011, Vander Weele and Shpitser, 2011, Wu et al., 2022]. This assumption is realistic as it reflects how samples are obtained in many application areas, such as economics and epidemiology [Hill, 2011, Imbens and Rubin, 2015, Wager and Athey, 2018]. For instance, every variable within the set $\mathbf{O}$ is measured prior to the implementation of the treatment and before the outcome is observed.*

---

[2]We present the notation for continuous random variables, with the corresponding discrete cases being straightforward.

[3]Note that this assumption can be checked in practice using observational data, as discussed in Section 5.

**Remark 2.** *Recently, Maasch et al. [2024] attempted to relax the pretreatment assumption and proposed the Local Discovery by Partitioning (LDP) method for identifying an adjustment set under a set of sufficient conditions. However, the method may fail to identify valid adjustment sets in certain cases where the EHS criterion succeeds, thereby limiting its completeness. For example, in the causal graph (c) of Figure 1, the LDP method fails to identify an adjustment set due to the violation of its sufficient condition, even though the pretreatment assumption holds. Furthermore, experimental results demonstrate that our proposed method remains effective on benchmark networks, even in scenarios where the pretreatment assumption is violated (see Section 4).*

**Task.** Under the standard assumptions of the causal Markov condition and the causal Faithfulness condition. Given an observational dataset $\mathcal{D}$ that consists of an ordered variable pair $(X, Y)$, along with a set of covariates $\mathbf{O}$, we focus on a local learning approach to tackle the challenge of determining whether a specific variable $X$ has a causal effect on another variable $Y$, allowing for latent variables in the system. If such a causal effect is present, we aim to locally identify an appropriate adjustment set of covariates that can provide a consistent and unbiased estimator of the true effect. Our method relies on analyzing the testable (conditional) independence and dependence relationships among the observed variables.

# 3 Local Search Adjustment Sets

In this section, we first present the identification results for the local search adjustment set. Based on these results, we then propose a local search algorithm for identifying the valid adjustment set and show that it is both sound and complete. All proofs are deferred to Appendix D for clarity.

## 3.1 Local Search Theoretical Results

In this section, we provide the theoretical results for estimating the unbiased causal effect $X$ on $Y$ (if such an effect exists) solely from the observational dataset $\mathcal{D}$. To this end, we need to locally identify the following three possible scenarios when the full causal structure is not known.

$\mathcal{S}1.$ $X$ has a causal effect on $Y$, and the causal effect is estimated by adjusting with a valid adjustment set.

$\mathcal{S}2.$ $X$ has no causal effect on $Y$.

$\mathcal{S}3.$ It is unknown whether there is a causal effect of $X$ on $Y$.

It should be emphasized that scenario $\mathcal{S}3$ arises because, under standard assumptions, based on the (testable) independence and dependence relationships among the observed variables, one may not identify a unique causal relationship between $X$ and $Y$. Typically, what we obtain is a Markov equivalence class encoding the same conditional independencies [Spirtes et al., 2000, Zhang, 2008b, Entner et al., 2013]. Thus, some of the causal relationships cannot be uniquely identified.[4]

We now address scenario $\mathcal{S}1$. Before that, we define the adjustment set relative to $(X, Y)$ within the Markov blanket of $Y$ in a MAG, denoted as $\mathcal{A}_{MB}(X, Y)$. This definition will help us locally identify a valid adjustment set using testable independencies and dependencies, even in the presence of latent variables.

**Definition 4** (Adjustment set in Markov blanket). *Let $(X, Y)$ be an ordered node pair in a MAG $\mathcal{M}$, where $\mathcal{M}$ is adjustment amenable w.r.t. $(X, Y)$. A set $\mathbf{Z}$ is an $\mathcal{A}_{MB}(X, Y)$ if and only if (1) $\mathbf{Z} \subseteq MB(Y) \setminus \{X\}$, (2) $\mathbf{Z} \cap Forb(X, Y) = \emptyset$, and (3) all non-causal paths from $X$ to $Y$ blocked by $\mathbf{Z}$.*

The intuition behind the concept of $\mathcal{A}_{MB}(X, Y)$ is as follows: in a graph without hidden variables, the causal effect of $X$ on $Y$ can be estimated using a subset of $Pa(Y) \setminus \{X\}$ [Pearl, 2009]. However, in practice, some nodes in $Pa(Y) \setminus \{X\}$ may be unobserved. For instance, consider the MAG shown in Figure. 1 (b), where the edge $V_4 \leftrightarrow Y$ indicates the presence of latent confounders. Consequently, the observed nodes do not include $Pa(Y)$. However, $MB(Y) = \{X, V_2, V_3, V_4\}$ contains the valid adjustment set $\{V_2, V_3, V_4\}$. According to Definition 4, we know $\{V_2, V_3, V_4\}$ is an $\mathcal{A}_{MB}(X, Y)$.

**Remark 3.** *Under our problem definition, since $\mathbf{O}$ is a set of pretreatment variables w.r.t. $(X, Y)$, we have $Forb(X, Y) = \{Y\}$ and $Y$ not in $MB(Y)$. Therefore, it is crucial to observe that the three*

---

[4]See Figure. 10 in Appdenix C.1 for an example.

*conditions in Definition 4 can be simplified to two: (1) $\mathbf{Z} \subseteq MB(Y) \setminus X$, and (2) all non-causal paths from $X$ to $Y$ are blocked by $\mathbf{Z}$.*

One may raise the following question: if no subset of $MB(Y) \setminus X$ qualifies as an adjustment set for $(X, Y)$, does it follow that no adjustment set for $(X, Y)$ exists within the covariate set $\mathbf{O}$? Interestingly, we find that the answer is yes, as formally stated in the following theorem.

**Theorem 1** (Existence of $\mathcal{A}_{MB}(X, Y)$). *Let $\mathcal{D}$ be an observational dataset containing an ordered variable pair $(X, Y)$ and a set of covariates $\mathbf{O}$. There exists a subset of $\mathbf{O}$ is an adjustment set w.r.t. $(X, Y)$ if and only if there exists a subset of $MB(Y) \setminus \{X\}$ is an adjustment set w.r.t. $(X, Y)$, i.e. , $\mathcal{A}_{MB}(X, Y)$.*

Theorem 1 states that if there exists a subset of $\mathbf{O}$ that is an adjustment set relative to $(X, Y)$, then there exists a subset of $MB(Y) \setminus \{X\}$ that is an adjustment set. Conversely, if no subset of $MB(Y) \setminus \{X\}$ is an adjustment set, then no subset of $\mathbf{O}$ is an adjustment set relative to $(X, Y)$.

**Example 2.** *Consider the MAG shown in Figure. 3 (b). We can observe $MB(Y)$ from the MAG, i.e. , $MB(Y) = \{X, V_1, V_2, V_3, V_4, V_6\}$. According to Definition 4 and the structure of the MAG, we can infer that any subset of $MB(Y) \setminus \{X\}$ that includes $\{V_1, V_2\}$ but excluding $\{V_3\}$ constitutes an $\mathcal{A}_{MB}(X, Y)$.*

Based on Theorem 1 and the rules in Entner et al. [2013], we next show that we can locally search $\mathcal{A}_{MB}(X, Y)$ by checking certain conditional independence and dependence relationships (Rule $\mathcal{R}1$), as stated in the following theorem. Meanwhile, we can locally find that $X$ has a causal effect on $Y$, i.e. , $\mathcal{S}1$.

**Theorem 2** ($\mathcal{R}1$ for Locally Searching Adjustment Sets). *Let $\mathcal{D}$ be an observational dataset containing an ordered variable pair $(X, Y)$ and a set of covariates $\mathbf{O}$. A subset $\mathbf{Z} \subseteq MB(Y) \setminus \{X\}$ is an $\mathcal{A}_{MB}(X, Y)$ if there exists a variable $S \in MB(X) \setminus \{Y\}$ such that (i) $S \not\perp\!\!\!\perp Y \mid \mathbf{Z}$, and (ii) $S \perp\!\!\!\perp Y \mid \mathbf{Z} \cup \{X\}$.*

Intuitively speaking, condition (i) indicates that there exist active paths from $S$ to $Y$ given $\mathbf{Z}$. Condition (ii) implies that there are no active paths from $S$ to $Y$ when given $\mathbf{Z} \cup \{X\}$. These two rules indicate that all active paths from $S$ to $Y$ given $\mathbf{Z}$ must pass through $X$. Thus, adding $X$ to the conditioning set blocks these active paths. Hence, all non-causal paths from $X$ to $Y$ are blocked by $\mathbf{Z}$; otherwise, condition (ii) will not hold. Then, according to Definition 4, we know $\mathbf{Z}$ is an $\mathcal{A}_{MB}(X, Y)$.

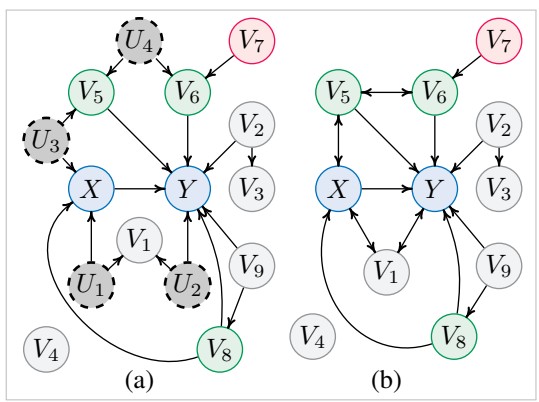

Figure 4: (a) An causal DAG, where $U_i, i = 1, ..., 4$ are latent variables. (b) The corresponding MAG of the DAG in (a).

**Example 3.** *Consider the causal diagram depicted in Figure. 4 (b). Assume that an oracle performs conditional independence tests on the observational dataset $\mathcal{D}$. Consequently, we can determine the $MB(X)$ and $MB(Y)$, i.e. , $MB(X) = \{V_1, V_2, V_5, V_6, V_7, V_8, V_9, Y\}$, and $MB(Y) = \{V_1, V_2, V_5, V_6, V_8, V_9, X\}$. According to Theorem 2, we can infer the existence of a causal effect of $X$ on $Y$. The set $\{V_5, V_6, V_8\}$ serves as an $\mathcal{A}_{MB}(X, Y)$, as $V_7 \not\perp\!\!\!\perp Y \mid \{V_5, V_6, V_8\}$ and $V_7 \perp\!\!\!\perp Y \mid \{V_5, V_6, V_8, X\}$.*

Next, we provide the rule $\mathcal{R}2$ that allows us to locally identify $X$ has no causal effect on $Y$, i.e. , $\mathcal{S}2$.

**Theorem 3** ($\mathcal{R}2$ for Locally Identifying No Causal effect). *Let $\mathcal{D}$ be an observational dataset containing an ordered variable pair $(X, Y)$ and a set of covariates $\mathbf{O}$. Then, $X$ has no causal effect on $Y$ if there exists a subset $\mathbf{Z} \subseteq MB(Y) \setminus \{X\}$ and a variable $S \in MB(X) \setminus \{Y\}$ such that at least one of the following conditions holds: (i) $X \perp\!\!\!\perp Y \mid \mathbf{Z}$, or (ii) $S \not\perp\!\!\!\perp X \mid \mathbf{Z}$ and $S \perp\!\!\!\perp Y \mid \mathbf{Z}$.*

According to the faithfulness assumption, condition (i) implies that $X$ and $Y$ are m-separated by a subset of $MB(Y) \setminus \{X\}$. Thus, $X$ has a zero effect on $Y$. Condition (ii) provides a strategy to identify a zero effect even when a latent confounder exists between $X$ and $Y$. Roughly speaking, $S \not\perp\!\!\!\perp X \mid \mathbf{Z}$ indicates that there are active paths from $S$ to $X$ given $\mathbf{Z}$. Therefore, if there were a

directed edge from $X$ to $Y$, it would create an active path from $S$ to $Y$ by connecting to the previous path, which would contradict condition $S \perp\!\!\!\perp Y \mid \mathbf{Z}$.

**Example 4.** *Consider the MAG shown in the Figure 3 (b). Assuming that the edge from $X$ to $Y$ is removed, then, we can infer that there is no causal effect of $X$ on $Y$ by condition (i), as $X \perp\!\!\!\perp Y \mid \{V_1, V_2\}$. Furthermore, suppose $V_2$ is a latent variable. Then, we can infer that there is no causal effect of $X$ to $Y$ by condition (ii), as $V_6 \not\perp\!\!\!\perp X \mid \{V_1\}$ and $V_6 \perp\!\!\!\perp Y \mid \{V_1\}$.*

Lastly, we show that if neither $\mathcal{R}1$ of Theorem 2 nor $\mathcal{R}2$ of Theorem 3 applies, then one cannot identify whether there is a causal effect of $X$ on $Y$, based on conditional independence and dependence relationships among the observational dataset $\mathcal{D}$, i.e. , we are in $\mathcal{S}3$.

**Theorem 4.** *Under the standard assumption, neither $\mathcal{R}1$ of Theorem 2 nor $\mathcal{R}2$ of Theorem 3 applies, then it is impossible to determine whether there is a causal effect of $X$ on $Y$, based on conditional independence and dependence relationships.*

Theorem 4 states that there may exist causal structures with and without an edge from $X$ to $Y$, that induce the same dependencies and independencies among the observational dataset $\mathcal{D}$. Consequently, it is not possible to uniquely infer whether there is a causal effect or not.

## 3.2 The LSAS Algorithm

In this section, we leverage the above theoretical results and propose the **L**ocal **S**earch **A**djustment **S**ets (LSAS) algorithm to infer whether there is a causal effect of a variable $X$ on another variable $Y$, and if so, to estimate the unbiased causal effect. Given an ordered variable pair $(X, Y)$, the algorithm consists of the following two key steps:

(i) **Learning the MBs of $X$ and $Y$**: This involves using an MB discovery algorithm to identify the Markov Blanket members (MBs) of both $X$ and $Y$.
(ii) **Determining Adjustment Sets:** For each variable $S$ in $MB(X) \setminus \{Y\}$, we check whether $S$ and the subsets $\mathbf{Z}$ of $MB(Y) \setminus \{X\}$ satisfy rules $\mathcal{R}1$ and $\mathcal{R}2$ based on Theorems $2 \sim 4$.

The algorithm uses $\Theta$ to store the estimated causal effect of $X$ on $Y$. If the output $\Theta$ is null, it suggests a lack of knowledge to obtain the unbiased causal effect, i.e. , $\mathcal{S}3$. If $\Theta = 0$, it indicates that there is no causal effect of $X$ on $Y$, i.e. , $\mathcal{S}2$. Otherwise, $\Theta$ provides the estimated causal effect of $X$ on $Y$, i.e. , $\mathcal{S}1$. The complete procedure is summarized in Algorithm 1, and the algorithm that we used for the MB learning is in Algorithm 2.

---

**Algorithm 1** Local Search Adjustment Sets (LSAS)

---

**Input:** Observational dataset $\mathcal{D}$, treatment variable $X$, outcome variable $Y$
1: $MB(X), MB(Y) \leftarrow$ Markov Blanket Discovery$(X, Y, \mathcal{D})$
2: $\Theta \leftarrow \emptyset$           // Initialize causal effect estimate
3: **for** each $S \in MB(X) \setminus \{Y\}$, each $\mathbf{Z} \subseteq MB(Y) \setminus \{X\}$ **do**
4:     **if** $S$ and $\mathbf{Z}$ satisfy $\mathcal{R}1$ (Theorem 2) **then**
5:        Estimate causal effect $\theta$ of $X$ on $Y$ given $\mathbf{Z}$, $\Theta \leftarrow \theta$.       // $\mathcal{S}1$
6:     **end if**
7:     **if** $S$ and $\mathbf{Z}$ satisfy $\mathcal{R}2$ (Theorem 3) **then**
8:        **return** $\Theta \leftarrow 0$       // No causal effect, i.e. , $\mathcal{S}2$
9:     **end if**
10: **end for**
**Output:** Estimated causal effect $\Theta$       // $\emptyset$, if scenario $\mathcal{S}3$ holds.

---

We next demonstrate that, in the large sample limit, the LSAS algorithm is both sound and complete.

**Theorem 5** (The Soundness and Completeness of LSAS Algorithm)**.** *Assume Oracle tests for conditional independence tests. Under the assumptions stated in our problem definition (Section 2.3), the LSAS algorithm correctly outputs the causal effect $\Theta$ whenever rule $\mathcal{R}1$ or $\mathcal{R}2$ applies. However, if neither rule $\mathcal{R}1$ nor $\mathcal{R}2$ applies, the LSAS algorithm can not determine whether there is a causal effect of $X$ on $Y$, based on the testable conditional independencies and dependencies among the observed variables.*

Formally, soundness means that, given an independence oracle and under the assumptions stated in our problem definition (Section 2.3), the inferences made using rule $\mathcal{R}1$ or $\mathcal{R}2$ are always correct

whenever these rules apply. On the other hand, completeness implies that if neither rule $\mathcal{R}1$ nor $\mathcal{R}2$ applies, it is impossible to determine, based solely on the conditional independencies and dependencies among the observed variables, whether $X$ has a causal effect on $Y$ or not.

**Complexity of Algorithm.** The LSAS algorithm's complexity comprises two main components:

1. MB discovery using the TC (Total Conditioning) algorithm [Pellet and Elisseeff, 2008b] with time complexity $\mathcal{O}(2n)$, where $n$ is the size of $\mathbf{O}$ plus $(X, Y)$, and
2. local identification using $\mathcal{R}_1$ and $\mathcal{R}_2$ (Lines $3 \sim 10$) with worst-case complexity $\mathcal{O}[(|MB(X)| - 1) \times 2^{|MB(Y)|-1}]$.

Thus, the overall worst-case time complexity is $\mathcal{O}[(|MB(X)| - 1) \times 2^{|MB(Y)|-1} + n]$. A detailed complexity analysis comparing LSAS with other algorithms is provided in Appendix C.3.

## 4 Experimental Results

To demonstrate the accuracy and efficiency of our proposed method, we applied it to synthetic data with random graphs, specific structures, and benchmark networks, as well as to the real-world dataset. We here use the existing implementation of the TC discovery algorithm [Pellet and Elisseeff, 2008b] to find the MB of a target variable. Our source code is available at *LSAS*.

**Comparison Methods.** We conducted a comparative analysis with several established techniques that do not require prior knowledge of the causal graph. Specifically, we evaluated our method against the LV-IDA with RFCI algorithm, which requires learning global graphs [Malinsky and Spirtes, 2016]; the EHS algorithm, which performs global searches under the pretreatment assumption without requiring graph learning [Entner et al., 2013]; the CEELS method, which conducts local searches under the pretreatment assumption [Cheng et al., 2022]; and the LDP method, which relaxes the pretreatment assumption for local searches [Maasch et al., 2024]. [5]

**Evaluation Metrics.** We evaluate the performance of the algorithms using the following typical metrics:

- **Relative Error (RE)**: the relative error of the estimated total causal effect ($\hat{CE}$) compared to the true total causal effect ($CE$), expressed as a percentage, formly,

$$RE = \left| \left( \hat{CE} - CE \right) / CE \right| \times 100\%$$

- **nTest**: the number of (conditional) independence tests implemented by an algorithm.

### 4.1 Synthetic Data

Following the conventions outlined in Malinsky and Spirtes [2016], Entner et al. [2013], we parameterized the random graphs, specific structures, and benchmark networks using a linear Gaussian causal model. The causal strength of each edge was sampled from a Uniform distribution $[0.5, 1.5]$, with additional noise terms drawn from a standard Gaussian distribution. Additionally, with linear regression, the causal effect of $X$ on $Y$ is calculated as the partial regression coefficient of $X$ [Malinsky and Spirtes, 2017]. Each experiment was repeated 100 times with randomly generated data, and the results were averaged. The sample sizes were set to 1K, 5K, 10K, and 15K, where K=1000. We based our experiments on the observed variables after removing the set of latent variables for each dataset. The experiments on random graphs are provided in Appendix E.1.

**Specific Structures.** We generated synthetic data based on the DAGs shown in Figure 3(a) and Figure 4(a). The corresponding MAGs, depicted in Figure 3(b) and 4(b), exclude the latent variables (e.g., $U_i$). The treatment variable is $X$ and the outcome variable is $Y$. Note that both DAGs exhibit M-structures [6]; adjusting for the collider $V_3$ leads to over-adjustment and introduces bias.

**Benchmark Networks.** Algorithms were evaluated on four benchmark Bayesian networks: *INSURANCE*, *MILDEW*, *WIN95PTS*, and *ANDES*. These networks contain 27 nodes with 52 arcs, 35 nodes

---

[5]Implementation sources: LV-IDA with RFCI algorithm (`https://github.com/dmalinsk/lv-ida`, using R-package pcalg [Kalisch et al., 2012]); EHS algorithm (`https://sites.google.com/site/dorisentner/publications/CovariateSelection`); and LDP method (`https://github.com/jmaasch/ldp`).

[6]The shape of the sub-graph looks like the capital letter M. See Figure 9 in Appendix B for more details.

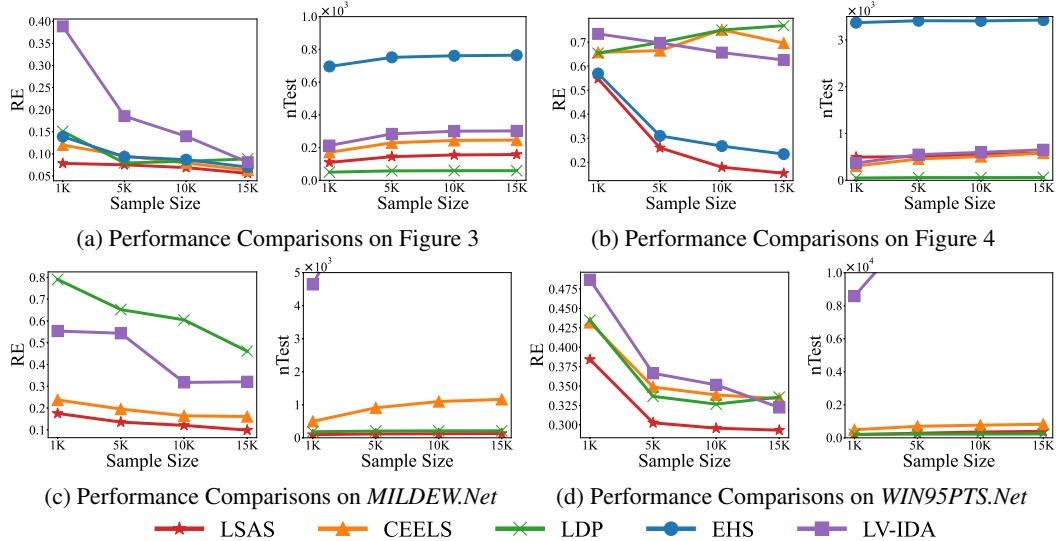

(a) Performance Comparisons on Figure 3

(b) Performance Comparisons on Figure 4

(c) Performance Comparisons on *MILDEW.Net*

(d) Performance Comparisons on *WIN95PTS.Net*

LSAS    CEELS    LDP    EHS    LV-IDA

Figure 5: Performance of five algorithms on *Specific Structures*, *MILDEW* and *WIN95PTS*.

with 46 arcs, 76 nodes with 112 arcs, and 223 nodes with 338 arcs, respectively.[7] The number of latent variables is set to 3, 5, 7, and 10 for the respective networks. In each dataset, we randomly selected latent variables and an ordered variable pair $(X, Y)$. Notably, the selected variable pairs may have descendants, which implies that the pretreatment assumption might not be satisfied.

**Results.** Due to space constraints, we present the results for *Specific Structures*, *MILDEW*, and *WIN95PTS* in Figure 5, while results for additional benchmark networks are provided in Appendix E.2. Note that some nTest values for LV-IDA are omitted from the plots as they exceed the scale limits. EHS results are not included in the benchmark networks plots due to excessive runtime. From these figures, we observe that our proposed LSAS algorithm outperforms other methods with almost all evaluation metrics in all structures and in all sample sizes. As expected, the nTest of our method is significantly lower than that of LV-IDA and EHS, which involve learning the global structure and globally searching for adjustment sets, respectively. Notably, in Figure 5 (b), CEELS and LDP show limited improvement with increasing sample size, which can be attributed to their lack of completeness in identifying valid adjustment sets. Additionally, LSAS outperforms other methods even when the pretreatment assumption may not be satisfied, as shown in Figure 5(c) and (d).

## 4.2 Real-world Dataset

In this section, we apply our method to a real-world dataset, the Cattaneo2 dataset, which contains birth weights of 4642 singleton births in Pennsylvania, USA Cattaneo [2010], Almond et al. [2005]. We here investigate the causal effect of a mother's smoking status during pregnancy $(X)$ on a baby's birth weight $(Y)$. The dataset [8] we used comprises 21 covariates, such as age, education, and health indicators for the mother and father, among others. Almond et al. [2005] have concluded that there is a strong negative effect of about $200 - 250$ g of maternal smoking $(X)$ on birth weight

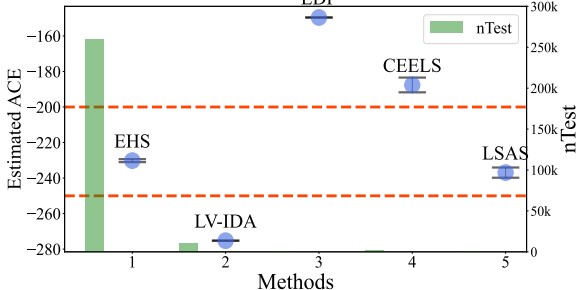

Figure 6: The causal effects and the number of (conditional) independence tests estimated by different methods, presented with 95% confidence intervals on the Cattaneo2 dataset. The two dotted lines represent the estimated interval provided in Almond et al. [2005].

---

[7]A detailed overview of these networks is provided in Table 3 in Appendix E.2. Additional information is available at `https://www.bnlearn.com/bnrepository/`.

[8]The dataset utilized in this study is available at `http://www.stata-press.com/data/r13/cattaneo2.dta`.

($Y$) using both subclassifications on the propensity score and regression-adjusted methods. Since there is no ground-truth causal graph and causal effects, we here use the negative effect of about $200 - 250$ as the baseline interval given in [Almond et al., 2005]. We follow Almond et al. [2005] to estimate the effect of maternal smoking on birth weight by regression-adjusted (see Section IV.C in [Almond et al., 2005]).

**Results.** The results of all methods are shown in Figure 6. It should be noted that due to the large number of nTest for EHS, the results for CEELS and LSAS are not clearly visible in the figure. In fact, the number of conditional independence tests for CEELS, LDP, and LSAS is 1284, 266, and 158, respectively. From the figure, we found that the effects estimated by EHS and LSAS fall within the baseline interval, while the effects estimated by other methods do not. Although the effect estimated by the EHS algorithm also falls within the baseline interval, LSAS requires fewer conditional independence tests, which means that LSAS is not only effective but also more efficient.

## 5    Limitations and Future Work

The preceding section presented how to locally search covariates solely from the observational data. Analogous to the setting studied by Entner et al. [2013] and Cheng et al. [2022], we assume that $Y$ is not a causal ancestor of $X$ and $X$ and $Y$ are not causal ancestors of any variables in **O** (pretreatment assumption). Regarding the first assumption, in practice, if one has no this prior knowledge, one can first use the existing local search structure algorithm allowing in the presence of latent variables, such as the MMB-by-MMB algorithm Xie et al. [2024], to identify whether $Y$ is not a causal ancestor of $X$. If it is, one can still use our proposed method to search for the adjustment set and estimate the causal effect. Regarding the pretreatment assumption, though many application areas can be obtained, such as economics and epidemiology Hill [2011], Imbens and Rubin [2015], Wager and Athey [2018], it may not always hold in real-world scenarios. Notably, existing methods—such as EHS, CEELS, and ours—do not directly extend to settings where the pretreatment assumption is violated, as they may select invalid adjustment sets that include descendants of the treatment, leading to biased estimates. For example, in the Figure 7,

if we choose $S = V_2$ and $\mathbf{Z} = \{V_4, V_6\}$, then $(S, \mathbf{Z})$ satisfies condition $R1$, but $\mathbf{Z}$ violates the generalized adjustment criterion since $V_6$ is a descendant of $X$. A potential workaround is to identify descendants of $X$ first, then apply $\mathcal{R}1$—but this lacks completeness, and may fail to recover adjustment sets even when they exist. To the best of our knowledge, no existing method can soundly and completely identify descendants of a treatment variable locally, especially in the presence of latent variables and without recovering the full graph. Addressing this remains an open and non-trivial challenge; it also deserves to explore methodologies that relax this assumption and address its violations[Maasch et al., 2024].

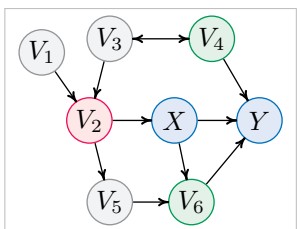

Figure 7: A MAG violating the pretreatment assumption with respect to $(X, Y)$.

Note that some causal effects cannot be identified only based on conditional independencies among observed data. Hence, leveraging background knowledge, such as data generation mechanisms Hoyer et al. [2008] and expert insights Fang and He [2020], to aid in identifying causal effects within local structures remains a promising research direction. In addition, obtaining data from multiple environments may also help identify the causal effect [Shi et al., 2021, De Bartolomeis et al., 2025].

## 6    Conclusion

We have introduced a novel local learning algorithm for covariate selection in nonparametric causal effect estimation with latent variables. Compared to existing methods, our approach does not require learning the global graph, is more efficient, and remains both sound and complete, even in the presence of latent variables.

## Acknowledgments and Disclosure of Funding

We appreciate the comments from anonymous reviewers, which greatly helped to improve the paper. This research was supported by the National Natural Science Foundation of China (62306019, 62472415). Feng Xie was supported by the Beijing Key Laboratory of Applied Statistics and Digital Regulation, and the BTBU Digital Business Platform Project by BMEC.

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

# Appendix Contents

# A  Related Work

This paper focuses on covariate selection in causal effect estimation within causal graphical models Pearl [2009], Spirtes et al. [2000]. Broadly speaking, the literature on covariate selection can be categorized into two main lines of research: methods that assume a known causal graph and methods that do not assume the availability of a causal graph. Below, we here provide a brief review of these two lines. For a comprehensive review of data-driven causal effect estimation, see Pearl [2009], Perkovi et al. [2018], Cheng et al. [2024].

**Methods with Known Causal Graph.** Ideally, when a causal graph is available, one can directly select an adjustment set for a causal relationship using the (generalized) back-door criterion Pearl [2009], Maathuis and Colombo [2015] or the (generalized) adjustment criterion Shpitser et al. [2010], Perkovi et al. [2018]. Research in this area often focuses on identifying special adjustment sets, such as minimal adjustment sets or 'optimal' valid adjustment sets that have the smallest asymptotic variance compared to other adjustment sets. For selecting minimal adjustment sets, see De Luna et al. [2011], Textor and Liśkiewicz [2011]. For 'optimal' valid adjustment sets, one may refer to Henckel et al. [2022] for semi-parametric estimators or Rotnitzky and Smucler [2020], Witte et al. [2020], Runge [2021] for non-parametric estimators. In contrast to the aforementioned methods, this paper focuses on the identification of valid adjustment sets under unknown causal graphs.

**Methods without Known Causal Graph.** A classical framework for inferring causal effect is IDA (Intervention do-calculus when the DAG is Absent) Maathuis et al. [2009]. IDA first learns a CPDAG and enumerates all Markov equivalent DAGs in the learned CPDAGs, then estimates all causal effects using the back-door criterion. Other notable developments along this line include combining prior knowledge Perkovic et al. [2017], Fang and He [2020] or employing strategies through local learning De Luna et al. [2011]. However, these methods often assume causal sufficiency, meaning no latent confounders exist in the system, and thus do not adequately account for the influences of latent variables. To address this limitation, a version of IDA suitable for systems with latent variables, known as LV-IDA (Latent Variable IDA), was proposed Malinsky and Spirtes [2017], based on the generalized back-door criterion Maathuis and Colombo [2015]. Subsequently, more efficient methods were proposed by Hyttinen et al. [2015], Wang et al. [2023], and Cheng et al. [2023]. Although these algorithms are effective, learning the global causal graph and estimating the causal effects for the entire system can be unnecessary and inefficient when the interest is solely on the causal effects of a single variable on an outcome variable.

To address this issue, Entner et al. [2013] proposed the EHS algorithm under the pretreatment assumption, demonstrating that the EHS method is both sound and complete for this task. However, the EHS approach is highly inefficient as it involves an exhaustive search over all possible combinations of variables for the inference rules. To overcome this inefficiency, Cheng et al. [2022] introduced a local algorithm called CEELS for selecting the adjustment set. While CEELS is faster than the EHS proposed by Entner et al. [2013], it may miss some adjustment sets during the local search that could be identified through a global search. In this paper, our work focuses on the same setting as EHS and introduces a fully local method for selecting the adjustment set. Compared to CEELS, our local method is both sound and complete, similar to the global learning method such as the EHS algorithm Entner et al. [2013]. More recently, to relax the pretreatment assumption, Maasch et al. [2024] proposed the Local Discovery by Partitioning (LDP) method, which identifies adjustment sets for exposure-outcome pairs under sufficient conditions. The LDP approach operates under less restrictive assumptions, and requires a quadratic number of conditional independence tests w.r.t. variable set size; however, it may fail to identify valid adjustment sets in cases where the EHS method is successful. In addition, Maasch et al. [2025] introduced the Local Discovery for Direct Discrimination (LD3) method, which targets the identification of adjustment sets for the weighted controlled direct effect, a specific type of causal effect. A notable limitation of LD3 is its requirement that all parents of the outcome variable be observed, a condition that may not be satisfied in some practical applications.

# B  More Details of Notations and Definitions

## B.1  Graph

A graph $\mathcal{G} = (\mathbf{V}, \mathbf{E})$ consists of a set of nodes $\mathbf{V} = \{V_1, \ldots, V_p\}$ and a set of edges $\mathbf{E}$. A graph $\mathcal{G}$ is ***directed mixed*** if the edges in the graph are *directed* ($\rightarrow$), or *bi-directed* ($\leftrightarrow$). The two ends of an edge

| Symbol | Description |
|--------|-------------|
| w.r.t. | With respect to |
| $(X, Y)$ | An ordered variable pair, where $X$ is the treatment and $Y$ is the outcome |
| $\mathcal{A}_{MB}(X, Y)$ | A valid adjustment set in $MB(Y) \setminus \{X\}$ w.r.t. $(X, Y)$ |
| $\mathcal{R}1$ | The rules in Theorem 2 |
| $\mathcal{R}2$ | The rules in Theorem 3 |
| $\mathbf{V}$ | The set of all variables, i.e., $\mathbf{V} = (X, Y) \cup \mathbf{O} \cup \mathbf{U}$ |
| $X$ | The treatment or exposure variable |
| $Y$ | The outcome or response variable |
| $\mathbf{O}$ | The set of observed covariates |
| $\mathbf{U}$ | The set of latent variables |
| $\mathcal{G}$ | A mix graph |
| $\mathcal{M}$ | A Maximal Ancestral Graph (MAG) |
| $\mathcal{P}$ | A Partial Ancestral Graph (PAG) |
| $t$ | The number of the observed covariates, i.e., $t = |\mathbf{O}|$ |
| $n$ | The number of the observed covariates plus the pair of nodes $(X, Y)$, i.e., $n = |(X, Y) \cup \mathbf{O}|$ |
| $Adj(V_i)$ | The set of adjacent nodes of $V_i$ |
| $MB(V_i)$ | The Markov blanket of a node $V_i$ in a MAG |
| $De(V_i)$ | The set of all descendants of $V_i$ |
| $PossDe(V_i)$ | The set of all possible descendants of $V_i$ |
| $(\mathbf{X} \perp\!\!\!\perp \mathbf{Y}|\mathbf{Z})_{\mathcal{G}}$ | A set $\mathbf{Z}$ m-separates $\mathbf{X}$ and $\mathbf{Y}$ in $\mathcal{G}$ |
| $\mathbf{X} \perp\!\!\!\perp \mathbf{Y}|\mathbf{Z}$ | $\mathbf{X}$ is statistically independent of $\mathbf{Y}$ given $\mathbf{Z}$. |
| $\mathbf{X} \not\perp\!\!\!\perp \mathbf{Y}|\mathbf{Z}$ | $\mathbf{X}$ is not statistically independent of $\mathbf{Y}$ given $\mathbf{Z}$ |
| $\mathcal{G}_X$ | The graph obtained from $\mathcal{G}$ by removing all visible directed edges out of $X$ in $\mathcal{G}$ |

Table 1: The list of main symbols used in this paper

are called **marks**. In a graph $\mathcal{G}$, two nodes are said to be **adjacent** in $\mathcal{G}$ if there is an edge (of any kind) between them. A node $V_i$ is a **parent**, **child**, or **spouse** of a node $V_j$ if there is $V_i \rightarrow V_j$, $V_i \leftarrow V_j$, or $V_i \leftrightarrow V_j$. A **path** $\pi$ in $\mathcal{G}$ is a sequence of distinct nodes $\langle V_0, \ldots, V_s \rangle$ such that for $0 \leq i \leq s - 1$, $V_i$ and $V_{i+1}$ are adjacent in $\mathcal{G}$. *The **length** of a path* equals the number of edges on the path. A **causal path** (directed path) from $V_i$ to $V_j$ is a path composed of directed edges pointing towards $V_j$, i.e. , $V_i \rightarrow \ldots \rightarrow V_j$. A **possibly causal path** (possibly directed path) from $V_i$ to $V_j$ is a path where every edge without an arrowhead at the mark near $V_i$. A path from $V_i$ to $V_j$ that is not possibly causal is called a **non-causal path** from $V_i$ to $V_j$, e.g. , $V_i \leftarrow V_{i+1} \leftarrow \ldots \rightarrow V_{j-1} \rightarrow V_j$. A path $\pi$ from $V_i$ to $V_j$ is a **collider path** if all the passing nodes are colliders on $\pi$, e.g. , $V_i \rightarrow V_{i+1} \leftrightarrow \ldots \leftrightarrow V_{j-1} \leftarrow V_j$. $V_i$ is called an **ancestor**, or **possible ancestor** of $V_j$ and $V_j$ is a **descendant**, or **possible descendant** of $V_i$ if there is a causal path, or possibly causal path from $V_i$ to $V_j$ or $V_i = V_j$. An **almost directed cycle** happens when $V_i$ is both a spouse and an ancestor of $V_j$. A **directed cycle** happens when $V_i$ is both a child and an ancestor of $V_j$.

**Definition 5** (**m-separation**). *In a directed mixed graph $\mathcal{G}$, a path $\pi$ between nodes $X$ and $Y$ is active (m-connecting) relative to a (possibly empty) set of nodes $\mathbf{Z}$ ($X, Y \notin \mathbf{Z}$) if 1) every non-collider on $\pi$ is not a member of $\mathbf{Z}$, and 2) every collider on $\pi$ has a descendant in $\mathbf{Z}$.*

A set $\mathbf{Z}$ *m-separates* $\mathbf{X}$ and $\mathbf{Y}$ in $\mathcal{G}$, denoted by $(\mathbf{X} \perp\!\!\!\perp \mathbf{Y}|\mathbf{Z})_{\mathcal{G}}$, if there is no active path between any nodes in $\mathbf{X}$ and any nodes in $\mathbf{Y}$ given $\mathbf{Z}$. The criterion of m-separation is a generalization of Pearl's d-separation criterion in DAG to ancestral graphs.

**Definition 6** (**Ancestral Graph and Maximal Ancestral Graph**). *A directed mixed graph is called an ancestral graph if the graph does not contain any directed or almost directed cycles (ancestral). In addition, an ancestral graph is a maximal ancestral graph (MAG) if for any two non-adjacent nodes, there is a set of nodes that m-separates them.*

**Definition 7** (**Markov Equivalence**). *Two MAGs $\mathcal{M}_1$, $\mathcal{M}_2$ are Markov equivalence if they share the same m-separations.*

Basically a Partial Ancestral Graph represents an equivalence class of MAGs.

**Definition 8** (**Causal Markov condition**). *The* causal markov condition *says the m-separation relations among the nodes in a graph $\mathcal{G}$ imply conditional independence in probability relations among the variables.*

**Definition 9** (**Causal Faithfulness condition**). *[Zhang, 2008a] The causal faithfulness condition states that m-connection in a graph $\mathcal{G}$ implies conditional dependence in the probability distribution.*

Under the above two conditions, conditional independence relations among the observed variables correspond exactly to m-separation in the MAG or PAG $\mathcal{G}$, i.e., $(\mathbf{X} \perp\!\!\!\perp \mathbf{Y}|\mathbf{Z})_P \Leftrightarrow (\mathbf{X} \perp\!\!\!\perp \mathbf{Y}|\mathbf{Z})_{\mathcal{G}}$.

**Definition 10** (**Partial Ancestral Graph**). *Zhang [2008b]] A Partial Ancestral Graph (PAG, denoted by $\mathcal{P}$) represents a $[\mathcal{M}]$, where a tail '$-$' or arrowhead '$>$' occurs if the corresponding mark is tail or arrowhead in all the Markov equivalent MAGs, and a circle '$\circ$' occurs otherwise.*

In other words, PAG contains all invariant arrowheads and tails in all the Markov equivalent MAGs. For convenience, we use an asterisk (*) to denote any possible mark of a PAG ($\circ, >, -$) or a MAG ($>, -$).

**Definition 11** (**Visible Edges**). *Zhang [2008a] Given a MAG $\mathcal{M}$ / PAG $\mathcal{P}$, a directed edge $X \to Y$ in $\mathcal{M}$ / $\mathcal{P}$ is **visible** if there is a node $S$ not adjacent to $Y$, such that there is an edge between $S$ and $X$ that is into $X$, or there is a collider path between $S$ and $X$ that is into $X$ and every non-endpoint node on the path is a parent of $Y$. Otherwise, $X \to Y$ is said to be **invisible**. Two possible configurations of the visible edge $X$ to $Y$ are provided as shown in Figure 8.*

A visible edge $X \to Y$ means that there are no latent confounders between $X$ and $Y$. All directed edges in DAGs and CPDAGs are said to be visible.

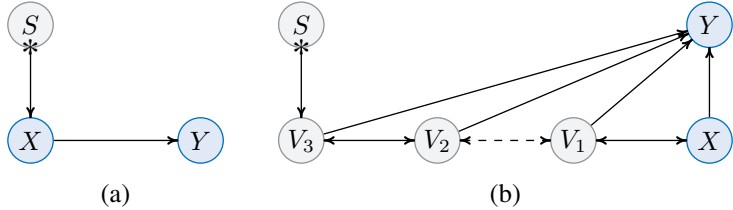

(a)  (b)

Figure 8: Two configurations where the edge $X \to Y$ is visible. Nodes $S$ and $Y$ must be nonadjacent in (a) and (b).

**Definition 12** ($\mathcal{G}_{\underline{X}}$ Maathuis and Colombo [2015]). *For a MAG $\mathcal{M}$, let $\mathcal{M}_{\underline{X}}$ denote the graph obtained from $\mathcal{M}$ by removing all visible directed edges out of $X$ in $\mathcal{M}$. For a PAG $\mathcal{P}$, let $\mathcal{M}$ be any MAG consistent with $\mathcal{P}$ that has the same number of edges into $X$ as $\mathcal{P}$, and let $\mathcal{P}_{\underline{X}}$ denote the graph obtained from $\mathcal{M}$ by removing all directed edges out of $X$ that are visible in $\mathcal{M}$.*

## B.2 Markov Blanket

**Definition 13** (**Markov Blanket**). *The Markov blanket of a variable $Y$, denoted as $MB(Y)$, is the smallest set conditioned on which all other variables are probabilistically independent of $Y$ [9], formally, $\forall V \in \mathbf{V} \setminus \{MB(Y) \cup V\} : Y \perp\!\!\!\perp V \mid MB(Y)$.*

Graphically, in a DAG, the Markov blanket of a node $Y$ includes the set of parents, children, and the parents of the children of $Y$. The Markov blanket of one node in a MAG is then defined as shown in Definition 14.

**Definition 14** (**MAG Markov Blanket** Richardson [2003], Pellet and Elisseeff [2008a], Yu et al. [2018]). *In a MAG $\mathcal{M}$, the Markov blanket of a node $Y$, noted as $MB(Y)$, comprises 1) the set of parents, children, and children's parents of $Y$; 2) the district of $Y$ and of the children of $Y$; and 3) the parents of each node of these districts. Where the district of a node $V$ is the set of all nodes reachable from $V$ using only bidirected edges.*

## B.3 M-structure

As shown in Figure 9 (a), the DAG is called M-structure (M-bias), where $U_1$ and $U_2$ are latent variables. This structure is very significant because it can lead to collider stratification bias, also

---

[9]Some authors use the term "Markov blanket" without the notion of minimality, and use "Markov boundary" to denote the smallest Markov blanket. For clarity, we adopt the convention that the *Markov blanket* refers to the minimal Markov blanket.

known as collider bias. The MAG corresponding to this DAG is shown in Fig. 9 (b). In this graph, according to the generalized adjustment criterion, if we are interested in the causal effect between $X$ and $Y$, we should not adjust for the variable $M$. Adjusting for $M$ would open the path $(X \leftrightarrow [M] \leftrightarrow Y)$, which was originally blocked. As a result, adjusting for the collider $M$ leads to over-adjustment and introduces bias.

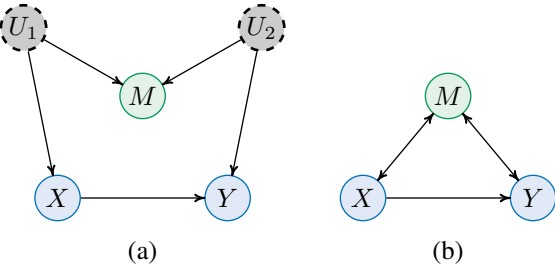

Figure 9: The illustrative example for M-structure. (a) A causal DAG, where $U_1$ and $U_2$ are latent variables. (b) The corresponding MAG of the DAG in (a).

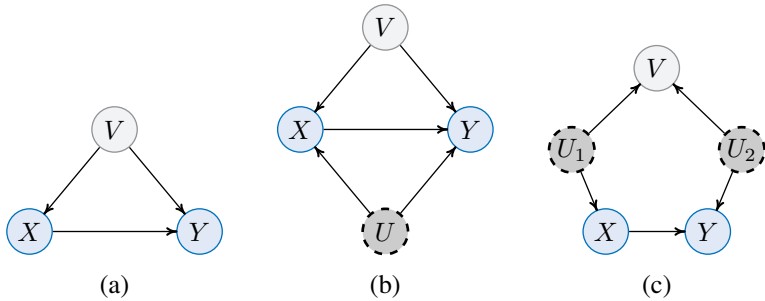

Figure 10: Three DAGs that entail the same independencies and dependencies among the observed variables $(X, Y, V)$, where $U_1$, $U_2$, and $U_3$ are latent variables.

## C    Supplement to Section 3

### C.1    An example for $\mathcal{S}3$

Consider the three graphs in Figure 10. These graphs entail the same independencies and dependencies among the observed variables $(X, Y, V)$. Consequently, it is impossible to determine, based solely on testable dependencies and independence, whether $X$ has a causal effect on $Y$ and whether $V$ should be included in the adjustment set.

### C.2    Markov Blanket Discovery Algorithm

In this section, we outline the procedure of the TC (Total Conditioning) algorithm Pellet and Elisseeff [2008b], which we used to discover the Markov blanket.

**Definition 15** (Total conditioning Pellet and Elisseeff [2008b]). *In the context of a faithful causal graph $\mathcal{G}$, we have:*

$$\forall X, Y \in \mathbf{V} : (X \in \text{Markov blanket}(Y)) \Leftrightarrow (X \not\perp\!\!\!\perp Y \mid \mathbf{V} \setminus \{X, Y\}) \tag{2}$$

### C.3    Complexity of Algorithms

The complexity of the LSAS algorithm can be divided into two main components: the first component is the MB discovery algorithm (Line 1), and the second involves locally identifying causal effects using $\mathcal{R}_1$ and $\mathcal{R}_2$ (Lines $3 \sim 10$). Let $n$ represent the size of the set $\mathbf{O}$ plus the ordered variable

**Algorithm 2** Markov Blanket Discovery Pellet and Elisseeff [2008b]

---

**Input:** Treatment variable $X$, outcome variable $Y$, observed covariates $\mathbf{O}$
1: Initialize $MB(X) \leftarrow \emptyset$, $MB(Y) \leftarrow \emptyset$
2: $\mathbf{V} = \mathbf{O} \cup \{X, Y\}$
3: **for** each $V_i \in \mathbf{O} \cup \{Y\}$ **do**
4:    **if** $X \not\perp V_i \mid \mathbf{V} \setminus \{X, V_i\}$ **then**
5:       Add $V_i$ to $MB(X)$                                 // Discover $MB(X)$
6:    **end if**
7: **end for**
8: **for** each $V_i \in \mathbf{O} \cup \{X\}$ **do**
9:    **if** $Y \not\perp V_i \mid \mathbf{V} \setminus \{Y, V_i\}$ **then**
10:      Add $V_i$ to $MB(Y)$                                 // Discover $MB(Y)$
11:    **end if**
12: **end for**
**Output:** $MB(X)$, $MB(Y)$

---

pair $(X, Y)$, and $|set|$ represents the size of the set. We utilized the TC (Total Conditioning) algorithm Pellet and Elisseeff [2008b] to identify the MB. Consequently, the time complexity of finding the MB for two nodes is $\mathcal{O}(2n - 3)$. In the worst-case scenario, the complexity of the second component is $\mathcal{O}[(|MB(X)| - 1) \times 2^{|MB(Y)|-1}]$. Therefore, the overall worst-case time complexity of the LSAS algorithm is $\mathcal{O}[(|MB(X)| - 1) \times 2^{|MB(Y)|-1} + n]$. Note that the complexity of the EHS algorithm is $\mathcal{O}[(n - 2) \times 2^{n-2}]$, which is significantly higher than the complexity of our algorithm, particularly when $n \gg |\text{MB}(Y)|$ in large causal networks. The CEELS algorithm Cheng et al. [2022] employs the PC.select algorithm Bühlmann et al. [2010] to search for $Adj(X)$ and $Adj(Y)$. In the worst-case scenario, the overall complexity of CEELS is $\mathcal{O}(n \times 2^q)$, where $q = \max(|Adj(X) \setminus Y|, |Adj(Y) \setminus X|)$. Although in practice, the complexity of CEELS may not differ significantly from that of our proposed algorithm, it is crucial to note that CEELS might miss an adjustment set during the local search that could otherwise be identified through a global search. This issue, as illustrated in Figure 1(b), is not present in our proposed algorithm. We list the complexity of the algorithms in Table 2.

Table 2: Summary of the algorithms features.

| Algorithm | Learning Graph | Time-complexity | Sound and Complete |
|-----------|:--------------:|:---------------:|:------------------:|
| LV-IDA | $\checkmark$ | $\mathcal{O}[n \times 2^{n-1}]$ Cheng et al. [2022], Malinsky and Spirtes [2016] | $\Leftrightarrow$ |
| EHS | $\times$ | $\mathcal{O}[(n - 2) \times 2^{n-2}]$ | $\Leftrightarrow$ |
| CEELS | $\times$ | $\mathcal{O}(n \times 2^q)$ | $\Rightarrow$ |
| LSAS | $\times$ | $\mathcal{O}[(|MB(X)| - 1) \times 2^{|MB(Y)|-1} + n]$ | $\Leftrightarrow$ |

Note: $\Rightarrow$ denotes sound, $\Leftrightarrow$ denotes sound and complete, and $\checkmark$ means graph learning is required; $\times$ means it is not.

# D Proofs

## D.1 Proof of Theorem 1

Before presenting the proof, we quote Theorem 1 of Xie et al. [2024] since it is used to prove Theorem 1.

**Lemma 1.** *[Theorem 1 of Xie et al. [2024]] Let $Y$ be any node in $\mathbf{O}$, and $X$ be a node in $MB(Y)$. Then $Y$ and $X$ are m-separated by a subset of $\mathbf{O} \setminus \{Y, X\}$ if and only if they are m-separated by a subset of $MB(Y) \setminus \{X\}$.*

Lemma 1 is an extended version of the result in Xie and Geng [2008], which considers the presence of latent variables. Notably, a fundamental fact is that: given any DAG $\mathcal{G}$ over $\mathbf{V} = \mathbf{O} \cup \mathbf{L}$—where $\mathbf{O}$ denotes the set of observed variables, and $\mathbf{L}$ denotes the set of latent variables—there is a MAG over $\mathbf{O}$ alone such that for any disjoint $\mathbf{X}, \mathbf{Y}, \mathbf{Z} \subseteq \mathbf{O}$, $\mathbf{X}$ and $\mathbf{Y}$ are d-separated by $\mathbf{Z}$ in $\mathcal{G}$ if and only if they are m-separated by $\mathbf{Z}$ in the MAG [Zhang, 2008a].

The intuitive implications of Lemma 1 are as follows: given a node $Y$ and another node $X$, where $X \in MB(Y)$, if there is a subset of $\mathbf{O} \setminus \{Y, X\}$ that m-separates $Y$ and $X$, then there must exist a subset of $MB(Y) \setminus \{X\}$ that m-separates $Y$ and $X$. Conversely, if no subset of $MB(Y) \setminus \{X\}$ m-separates $Y$ and $X$, then no subset of $\mathbf{O} \setminus \{Y, X\}$ can m-separate $Y$ and $X$.

We now proceed to establish the proof of Theorem 1.

*Proof.* According to Definition 3, a set $\mathbf{Z}$ is a valid adjustment set w.r.t. $(X, Y)$ in $\mathcal{P}$ if it satisfies all the conditions therein. When the treatment $X$ is a singleton, the generalized adjustment criterion becomes equivalent to the *generalized back-door criterion* proposed by Maathuis and Colombo [2015]. Under our problem definition, the above conditions can be simplified: a set $\mathbf{Z} \subseteq \mathbf{O}$ is a valid adjustment set w.r.t. $(X, Y)$ if $\mathcal{P}$ is adjustment amenable relative to $(X, Y)$ (i.e., $X$ and $Y$ are connected by a visible edge, as a visible $X \to Y$) and $\mathbf{Z}$ m-separates $X$ and $Y$ in the $\mathcal{P}_{\underline{X}}$.

Equivalently, this is to show that a subset of $\mathbf{O}$ is an m-separating set w.r.t. $(X, Y)$ in $\mathcal{P}_{\underline{X}}$ if and only if a subset of $MB'(Y)$ is an m-separating set w.r.t. $(X, Y)$ in $\mathcal{P}_{\underline{X}}$, where $MB'(Y)$ denotes the MB of $Y$ in $\mathcal{P}_{\underline{X}}$. Note that $MB'(Y) \subseteq MB(Y)$, and $X$ may not belong to $MB'(Y)$ in $\mathcal{P}_{\underline{X}}$. We now analyze two cases:

**Case 1:** Suppose $\mathcal{P}$ is adjustment amenable relative to $(X, Y)$, with $X \in MB'(Y)$ in $\mathcal{P}_{\underline{X}}$, it follows that $X \notin Adj(Y)$. According to Lemma 1 and the fact that $X \in MB'(Y)$ in $\mathcal{P}_{\underline{X}}$, $X$ and $Y$ are m-separated by a subset of $\mathbf{O}$ if they are m-separated by a subset of $MB'(Y) \setminus \{X\}$ in $\mathcal{P}_{\underline{X}}$. If no subset of $MB'(Y) \setminus \{X\}$ m-separates $X$ and $Y$, then no subset of $\mathbf{O} \setminus \{X, Y\}$ can m-separate $X$ and $Y$, which implies $X \in Adj(Y)$ in $\mathcal{P}_{\underline{X}}$. This contradicts the assumption, thus proving that $\mathcal{P}$ is not adjustment amenable relative to $(X, Y)$.

**Case 2:** Suppose $\mathcal{P}$ is adjustment amenable relative to $(X, Y)$, with $X \notin MB'(Y)$ in $\mathcal{P}_{\underline{X}}$, it follows that $X \notin Adj(Y)$. Thus, $X$ and $Y$ are m-separated by $MB'(Y)$, i.e., $(X \perp\!\!\!\perp Y \mid MB'(Y))_{\mathcal{P}_{\underline{X}}}$. If $(X \not\perp\!\!\!\perp Y \mid MB'(Y))_{\mathcal{P}_{\underline{X}}}$, this contradicts the assumption, showing $\mathcal{P}$ is not adjustment amenable relative to $(X, Y)$. Consequently, no subset of $\mathbf{O} \setminus \{X\}$ is a valid adjustment set w.r.t. $(X, Y)$ in $\mathcal{P}$. $\square$

## D.2 Proof of Theorem 2

*Proof.* By $S \not\perp\!\!\!\perp Y \mid \mathbf{Z}$, we can be certain that there exist active paths from $S$ to $Y$ given $\mathbf{Z}$. In addition, $S \perp\!\!\!\perp Y \mid \mathbf{Z} \cup \{X\}$ ensures that all such active paths must go through $X$, as all paths are blocked by adding $X$ to the conditioning set.

According to the pretreatment assumption, $X$ is not a causal ancestor of any nodes in $\mathbf{O}$, and $X$ is not included in the conditioning set for condition (i). Hence, all active paths from $S$ to $Y$ given $\mathbf{Z}$ must include a direct edge from $X$ to $Y$. Otherwise, if $X$ is a collider, then $X$ would need to be in the conditioning set for the active paths from $S$ to $Y$. Therefore, these two conditions determine that $X$ has a causal effect on $Y$.

From $S \not\perp\!\!\!\perp Y \mid \mathbf{Z}$ and $S \perp\!\!\!\perp Y \mid \mathbf{Z} \cup \{X\}$, it follows that there exists at least one active path from $S$ to $X$ given $\mathbf{Z}$, pointing into $X$ (by the pretreatment assumption). Suppose there exists an active non-causal path between $X$ and $Y$ given $\mathbf{Z}$. By connecting these paths, we obtain an active path from $S$ to $Y$ given $\mathbf{Z} \cup \{X\}$ (by Lemma 3.3.1 of Spirtes et al. [2000], when these paths share more than one node), due to a collider at $X$. That is, $S \not\perp\!\!\!\perp Y \mid \mathbf{Z} \cup \{X\}$. However, this contradicts $S \perp\!\!\!\perp Y \mid \mathbf{Z} \cup \{X\}$. Therefore, such active non-causal paths do not exist, and $\mathbf{Z}$ must block all non-causal paths from $X$ to $Y$, i.e., $\mathbf{Z}$ is a $\mathcal{A}_{MB}(X, Y)$.

$\square$

## D.3 Proof of Theorem 3

*Proof.* Under faithfulness, condition (i) infers that there is no edge between $X$ and $Y$, and $X$ is not a causal ancestor of any nodes in $\mathbf{O}$, implying there is no causal path from $X$ to $Y$ unless there is a directed edge from $X$ to $Y$. Hence, there is no causal effect of $X$ on $Y$.

Condition (ii) implies that $X$ and $Y$ are connected through a latent confounder. The condition $S \not\perp\!\!\!\perp X \mid \mathbf{Z}$ ensures that there are active paths from $S$ to $X$ given $\mathbf{Z}$, and since $X$ is not a causal ancestor of any nodes in $\mathbf{O}$, these active paths must point to $X$. Since $Y \notin \mathbf{Z}$ and $Y$ is not a causal

ancestor of any nodes in $\mathbf{O}$ or $X$, these active paths do not pass through $Y$. If there were a directed edge from $X$ to $Y$, it would create active paths from $S$ to $Y$, contradicting the condition $S \perp\!\!\!\perp Y \mid \mathbf{Z}$. Thus, there is no causal effect of $X$ on $Y$. $\qquad\square$

### D.4 Proof of Theorem 4

To prove Theorem 4, we need to analyze two scenarios:

**Scenario 1:** First, we will prove $\mathcal{R}1$ that in Theorem 2 is a necessary condition for identifying the existence of a causal effect of $X$ on $Y$ that can be inferred from $\mathcal{D}$ and a set of variables $\mathbf{Z}$ is $\mathcal{A}_{MB}(X, Y)$. That is, if there exists the causal effect of $X$ on $Y$ that can be inferred through conditional independence tests from $\mathcal{D}$, then a set $\mathbf{Z}$ is $\mathcal{A}_{MB}(X, Y)$, and $S \not\perp\!\!\!\perp Y \mid \mathbf{Z}$ and $S \perp\!\!\!\perp Y \mid \mathbf{Z} \cup \{X\}$, where $S \in MB(X) \setminus \{Y\}$ and $\mathbf{Z} \subseteq MB(Y) \setminus \{X\}$.

Suppose we are given the causal PAG $\mathcal{P}$ that was learned from testable (conditional) independence and dependence relationships among the observed variables. Consider the generalized adjustment criterion in Perkovi et al. [2018], which is a sound and complete graphical criterion for covariate adjustment in DAGs, CPDAGs, MAGs, and PAGs. According to the amenability condition, there is a visible edge $X \to Y$ in $\mathcal{P}$. According to the definition of visible edges: a directed edge $X \to Y$ in $\mathcal{P}$ is visible if there is a node $V$ not adjacent to $Y$, such that:

1. There is an edge between $V$ and $X$ that is into $X$, or
2. There is a collider path between $V$ and $X$ that is into $X$ and every non-endpoint node on the path is a parent of $Y$.

Otherwise, $X \to Y$ is invisible. Treating such a node $V$ as an $S$, we then consider it in two cases.

- **Case 1:** There exists an $S$ in $\mathcal{P}$ that satisfies the above case (1), i.e., there is an edge between $S$ and $X$ that points to $X$, and $S$ is not adjacent to $Y$. In other words, there exists a path that $S *\!\to X \to Y$ in the $\mathcal{G}$. Consequently, in this case, $S \not\perp\!\!\!\perp Y \mid \mathbf{Z}$ always holds because $X$ is not included in the conditioning set. According to Theorem 1, there is at least a set $\mathbf{Z} \subseteq MB(Y) \setminus \{X\}$ is $\mathcal{A}_{MB}(X, Y)$, meaning $\mathbf{Z}$ block all non-causal paths from $X$ to $Y$. Adding $X$ to the condition set will block the path $S *\!\to X \to Y$, and the non-causal paths from $X$ to $Y$ are blocked by $\mathbf{Z}$. Therefore, $S \perp\!\!\!\perp Y \mid \mathbf{Z} \cup \{X\}$ holds.

- **Case 2:** If there exists an $S$ in $\mathcal{P}$ that satisfies the above case (2), i.e., there is a collider path between $S$ and $X$ that is into $X$ and every non-endpoint node on the path is a parent of $Y$, and $S$ not adjacent to $Y$. In this case, these collider nodes all belong to $MB(X)$ and $MB(Y)$. Assuming that there is no active path from $S$ to $X$, placing these collider nodes into the condition set will activate this $S$ to $X$ collider path. In addition, these collider nodes must be in the condition set $\mathbf{Z}$ for block non-causal paths that pass these nodes. Thus, $S \not\perp\!\!\!\perp X \mid \mathbf{Z}$ will hold. According to Theorem 1, a set $\mathbf{Z}$ blocks all non-causal paths from $X$ to $Y$. The newly activated path between $S$ and $X$ and the path after the $X \to Y$ merger are not blocked by $\mathbf{Z}$. Adding $X$ to the conditional set would then block this path, and thus $S \perp\!\!\!\perp Y \mid \mathbf{Z} \cup \{X\}$ holds.

In summary, these two cases prove that $\mathcal{R}1$ is a necessary condition for identifying the causal effect of $X$ on $Y$ that can be inferred through conditional independence tests.

**Scenario 2:** Second, we will prove that $\mathcal{R}2$ in Theorem 3 is a necessary condition for identifying the absence of the causal effect of $X$ on $Y$ that can be inferred from the testable (conditional) independence and dependence relationships among the observed variables. That is, if the absence of the causal effect of $X$ on $Y$ that can be inferred from $\mathcal{D}$, then $X \perp\!\!\!\perp Y \mid \mathbf{Z}$, or $S \not\perp\!\!\!\perp X \mid \mathbf{Z}$ and $S \perp\!\!\!\perp Y \mid \mathbf{Z}$, where $S \in MB(X) \setminus \{Y\}$ and $\mathbf{Z} \subseteq MB(Y) \setminus \{X\}$.

Assuming Oracle tests for conditional independence tests. Under our problem definition, the causal structures discovered through testable conditional independence and dependencies between observable variables, which can infer $X$ has no causal effect on $Y$ can be divided into the following two cases:

1. There is no edge between $X$ and $Y$.
2. The edge between $X$ and $Y$ is $X \leftrightarrow Y$.

We then consider the two cases separately.

- **Case 1:** If there is no edge between $X$ and $Y$, then by Lemma 1, if $X \in MB(Y)$, there must exist a subset $\mathbf{Z}$ of $MB(Y) \setminus \{X\}$ that m-separates $X$ and $Y$, i.e., $X \perp\!\!\!\perp Y \mid \mathbf{Z}$, where $\mathbf{Z} \subseteq MB(Y) \setminus \{X\}$. If $X \notin MB(Y)$, then by Definition 13, $X \perp\!\!\!\perp Y \mid MB(Y)$.

- **Case 2:** Since $Y$ and $X$ are not causal ancestors of any nodes in $\mathbf{O}$, and $Y$ is not a causal ancestor of $X$, then $X$ is a collider. If $X$ is an unshielded collider, then there exists a node $S$ that is adjacent to $X$, but not to $Y$. Such $S$ belong to $MB(X) \setminus \{Y\}$ and $MB(Y) \setminus \{X\}$. Then by Lemma 1, $S \perp\!\!\!\perp Y \mid \mathbf{Z}$, where $\mathbf{Z} \subseteq MB(Y) \setminus \{X\}$. In addition, $S \not\perp\!\!\!\perp X \mid \mathbf{Z}$ due to $S$ is adjacent to $X$. If $X$ is a shielded collider, which can be inferred by testable conditional independence and dependencies between observable variables, then there exists a discriminating path $p$ for $X$ Zhang [2008b]. This path $p$ includes at least three edges, $X$ is a non-endpoint node on $p$ and is adjacent to $Y$ on $p$. The path has a node $S$ that is not adjacent to $Y$, and every node between $S$ and $X$ on $p$ is a collider and a parent of $Y$. The colliders between $S$ and $X$ belong to both $MB(X)$ and $MB(Y)$. Including these collider nodes in the set $\mathbf{Z}$ ensures that $S \perp\!\!\!\perp Y \mid \mathbf{Z}$, where $\mathbf{Z}$ consists of nodes from $MB(Y)$ that m-separate $S$ and $Y$. Furthermore, $S \not\perp\!\!\!\perp X \mid \mathbf{Z}$ holds because $\mathbf{Z}$ includes these colliders between $X$ and $S$.

*Proof.* Based on the above analysis, if $\mathcal{R}1$ does not apply, then we cannot infer whether there is a causal effect of $X$ on $Y$ from the independence and dependence relationships among the observed variables. If $\mathcal{R}2$ does not apply, then we cannot infer that there is no causal effect of $X$ on $Y$ from the independence and dependence relationships among the observed variables. Consequently, if neither $\mathcal{R}1$ nor $\mathcal{R}2$ applies, then we cannot infer whether there is a causal effect of $X$ on $Y$ from the independence and dependence relationships among the observed variables. $\square$

### D.5 Proof of Theorem 5

*Proof.* Assuming Oracle tests for conditional independence tests, the MB discovery algorithm finds all and only the MB nodes of a target variable.

Following Theorem 2 and Theorem 4, $\mathcal{R}1$ is a sufficient and necessary condition for identifying $X$ has a causal effect on $Y$ that can be inferred by testable (conditional) independence and dependence relationships among the observational variables, and there is a set $\mathbf{Z}$ is $\mathcal{A}_{MB}(X, Y)$. Hence, if there is a causal effect of $X$ on $Y$ that can be inferred by observational data, then LSAS can accurately identify the causal effect of $X$ on $Y$.

Subsequently, relying on Theorem 3 and Theorem 4, $\mathcal{R}2$ is a sufficient and necessary condition for identifying the absence of the causal effect of $X$ on $Y$ that can be inferred from observational data. Thus, LSAS can correctly identify that there is no causal effect of $X$ on $Y$ that can be inferred from observational data. Ultimately, if neither $\mathcal{R}1$ nor $\mathcal{R}2$ applies, then LSAS cannot infer whether there is a causal effect of $X$ on $Y$ from the independence and dependence relationships between the observations.

Hence, the soundness and completeness of the LSAS algorithm are proven. $\square$

## E More Results on Experiments

All experiments were conducted on an Intel CPU running at 3.60 GHz, with 64 GB of memory. For all methods, the significance level for the individual conditional independence tests is set to 0.01. The maximum size of the conditioning sets considered is 3 for *Specific Structures*, 5 for the *INSURANCE* and *MILDEW* networks, and 7 for the *WIN95PTS* and *ANDES* networks.

### E.1 Experimental Results on Random Graphs

### E.1.1 Experimental Results with Varying Numbers of Nodes

We conducted experiments on random graphs generated using the Erdős-Rényi model $G(n, d)$ [Erd6s and Rényi, 1960], where $n$ represents the number of nodes and $d$ denotes the average degree of each node. In our experiments, we set the number of nodes to 20, 30, 40, and 50, respectively, with an average degree of 3 for each node. The last two variables in the causal ordering were designated as the ordered variable pair $(X, Y)$. Additionally, $10\% \times n$ nodes with two or more children were randomly selected as unobserved variables in each experiment.

**Results.** The experimental results on random graphs are presented in Figure 11. For clarity, some nTest values for LV-IDA are omitted from the plots as they exceed the scale limits. Similarly, the nTest values for EHS in Figure 11(a) are excluded for the same reason. Additionally, when $n > 20$, EHS results are not shown due to excessive runtime. The results demonstrate that our proposed LSAS algorithm achieves superior performance compared to other methods across almost all evaluation metrics, network structures, and sample sizes. This superior performance validates the effectiveness and efficiency of LSAS across networks with varying numbers of nodes.

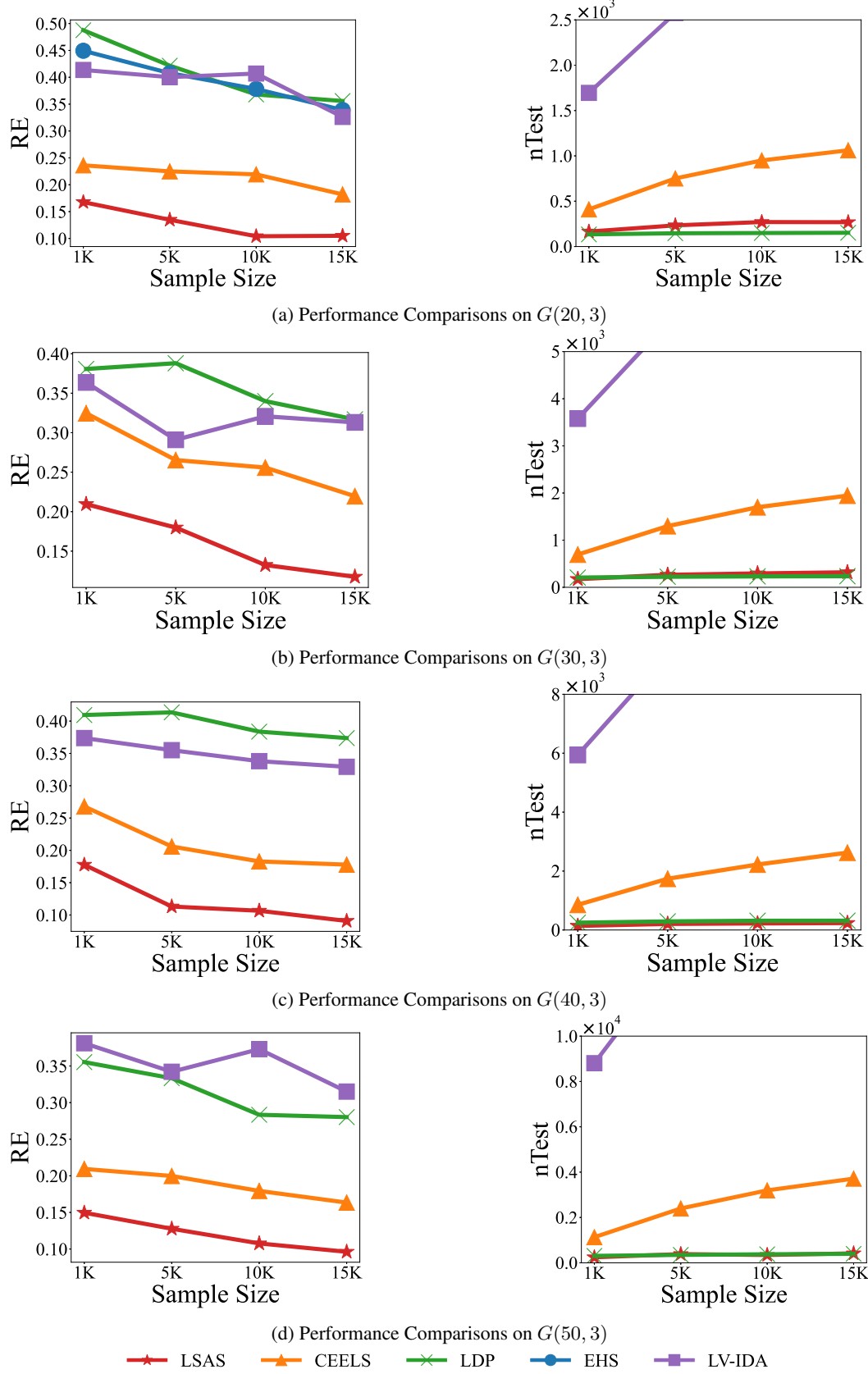

(a) Performance Comparisons on $G(20, 3)$

(b) Performance Comparisons on $G(30, 3)$

(c) Performance Comparisons on $G(40, 3)$

(d) Performance Comparisons on $G(50, 3)$

Figure 11: Performance of five algorithms on random graphs

### E.1.2 Experimental Results with Varying Average Degree

In this section, we conducted additional experiments on random graphs with varying average degrees (3, 5, 7, and 9) under the same settings as in Appendix E.1.1 (40 nodes, sample size = 5K). Due to excessive computational cost, the global method EHS failed to terminate within 2 hours on these denser graphs. To ensure a fair comparison, we here report its performance based on an early stopping threshold of 200 seconds.

**Results.** As shown in Figure 12, LSAS consistently outperforms all baseline methods in terms of RE and nTests, even as graph density increases. The only exception occurs in the $(40, 9)$ network, where the local method LDP achieves a lower nTest, while LSAS attains a substantially lower RE. Although the performance of all methods declines with increasing density—as expected—LSAS preserves a clear advantage, particularly in RE accuracy.

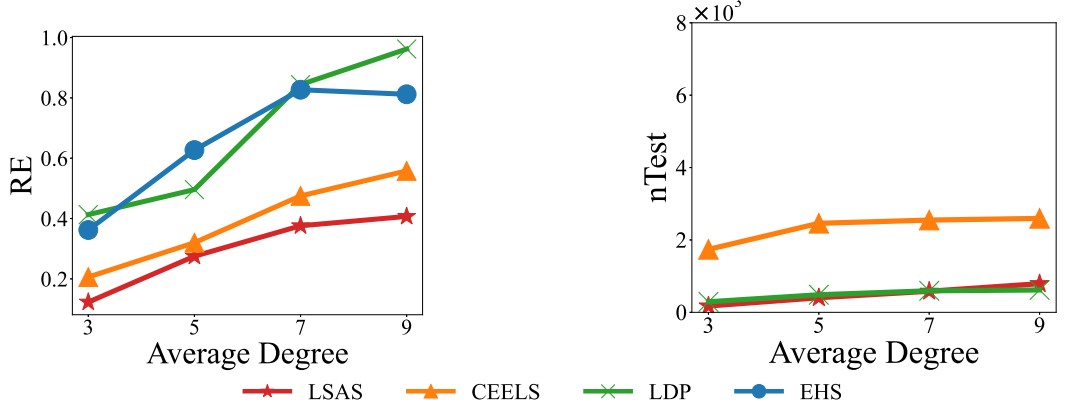

Figure 12: Performance of four algorithms on random graphs with varying average degrees

### E.2 Experimental Results on Benchmark Networks

### E.2.1 Overview of Benchmark Networks

Table 3 provides a detailed overview of the benchmark network statistics used in this paper. "Max in-degree" refers to the maximum number of edges pointing to a single node, while "Avg degree" denotes the average degree of all nodes.

Table 3: Statistics on the Networks.

| Networks | Num.nodes | Number of arcs | Max in-degree | Avg degree |
|---|---|---|---|---|
| *INSURANCE* | 27 | 52 | 3 | 3.85 |
| *MILDEW* | 35 | 46 | 3 | 2.63 |
| *WIN95PTS* | 76 | 112 | 7 | 2.95 |
| *ANDES* | 223 | 338 | 6 | 3.03 |

### E.2.2 Experimental Results on the *INSURANCE* and *ANDES* Benchmark Networks

Experimental results on the *INSURANCE* and *ANDES* benchmark Bayesian networks are provided in Figure 13.

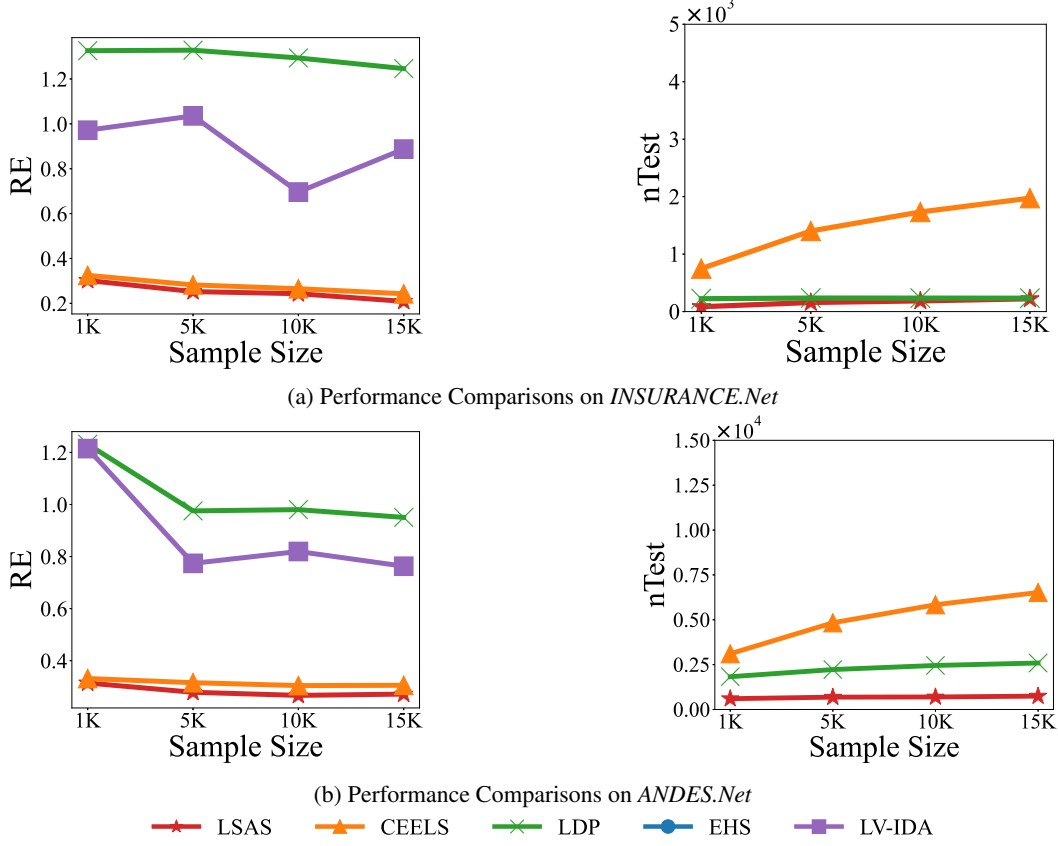

(a) Performance Comparisons on *INSURANCE.Net*

(b) Performance Comparisons on *ANDES.Net*

★ LSAS    ▲ CEELS    ✕ LDP    ● EHS    ■ LV-IDA

Figure 13: Performance of five algorithms on *INSURANCE* and *ANDES*

### E.2.3 Runtime Performance Comparison

In this section, we additionally report the runtime of the algorithms on the benchmark Bayesian networks, as shown in Figure 14. LSAS consistently outperforms baselines in runtime across most graphs and sample sizes. One exception is the WIN95PTS network at large sample sizes (15K), where the local method LDP is faster; however, LSAS achieves substantially higher RE accuracy (see Figures 5 and 13), due to LDP's incompleteness in identifying valid adjustment sets.

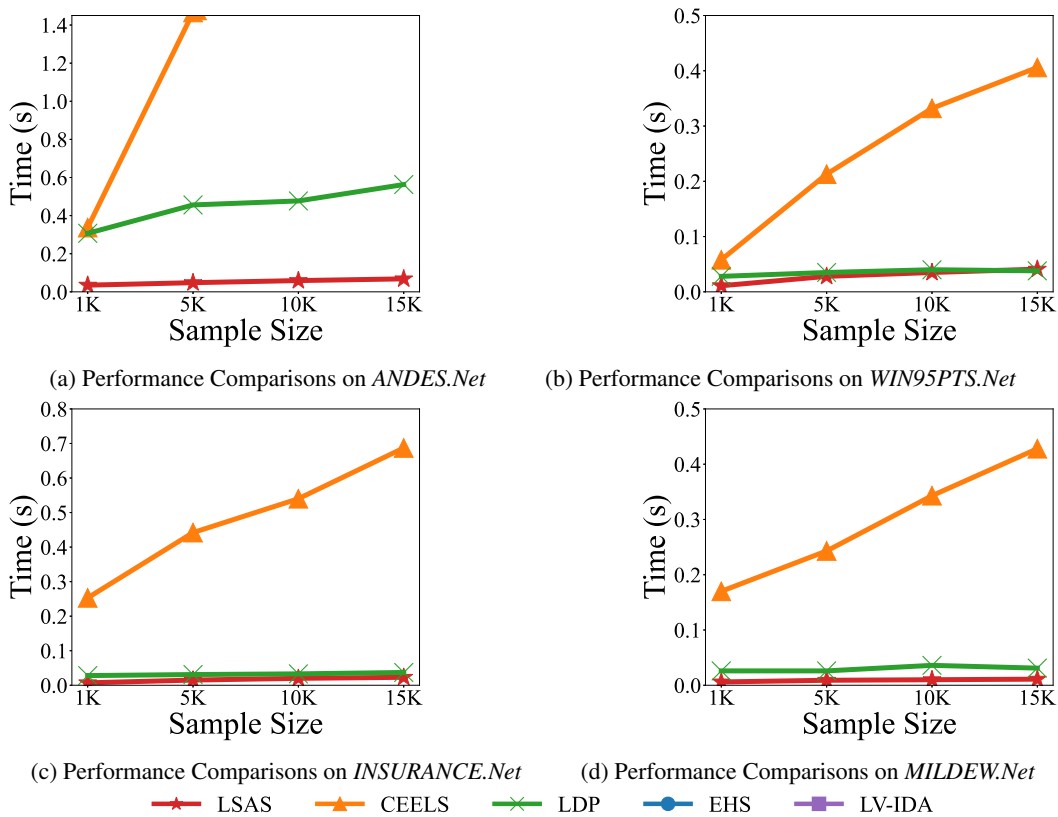

Figure 14: Performance of five algorithms on benchmark networks

