# OpenReview forum: "Local Learning for Covariate Selection in Nonparametric Causal Effect Estimation with Latent Variables"
_NeurIPS.cc/2025/Conference — NeurIPS 2025 poster_

### Official Review · Reviewer_swW2 · 2025-06-17

**Clarity:** 4
**Significance:** 3
**Originality:** 3
**Rating:** 5
**Confidence:** 4

**Summary:**

Causal effect estimation is a fundamental problem in many domains, such as epidemiology, economics, and political science. One popular method for causal effect estimation is known as backdoor adjustment. The challenge with backdoor adjustment, however, is that it may not always be clear which variables constitute a valid backdoor adjustment set. The authors propose a new method known as LSAS that improves upon known methods in the literature, most notably in time complexity. Through synthetic simulations and an analysis on a real-world dataset, the authors demonstrate the effectiveness and improvements of their method.

**Questions:**

In your definition for a maximal ancestral graph, is that the same as an acyclic directed mixed graph (ADMG)? Further, what is an “almost directed cycle” in lines 90-91?

Theorems 2,3, and 4 are nearly exactly the same as the inference rules and Theorems 2 and 3 presented in Entner et al. (2013). Is there a specific reason why the authors did not cite Entner et al. (2013) when presenting these three theorems? It seems to me that the proofs of Theorems 2, 3, and 4 should follow in a fairly straightforward manner from the proofs in Entner et al. (2013) once Theorem 1 is established. I believe that it is important to cite Entner et al. (2013) when presenting these three theorems as they are, at the very least, heavily inspired by the work of EHS.

Is faithfulness implicit in the theorems of this paper? Line 257 is the first mention of the faithfulness assumption in the main paper. If the authors are assuming faithfulness throughout, then they should define faithfulness and state the assumption somewhere in Section 2.

**Ethical Concerns:**

["NO or VERY MINOR ethics concerns only"]

**Final Justification:**

This is a well-written paper with non-trivial theoretical results that improve upon methods in the current literature. The authors show that it is possible to significantly reduce the search space of the problem without sacrificing soundness and completeness. My concerns regarding the paper only had to do with the writing style, and they have all been addressed. Therefore, I recommend acceptance of this paper.

**Limitations:**

Yes.

**Paper Formatting Concerns:**

No formatting concerns.

**Quality:**

4

**Strengths And Weaknesses:**

Strengths

A major strength of this paper is its clarity of writing. The literature review is comprehensive, and it is easy to follow the authors’ claims and their examples. All major claims are supported by theoretical proofs and simulation studies.

The key insight of this paper is that, for the purposes of learning a valid covariate adjustment set, one only needs to look at the Markov blankets of $X$ and $Y$ because, as the authors prove, there is a valid adjustment set amongst all of the observed variables if and only if there is a valid adjustment set amongst the Markov blankets of $X$ and $Y$. When the number of potential confounders is large, reducing the search space to just the Markov blankets can result in huge improvements in efficiency. This insight is, to the best of my knowledge, not previously known in the literature. The rest of the paper then uses this insight and known methods, most notably an efficient algorithm for learning the Markov blanket of a variable, to develop their method and algorithm.

The simulation studies clearly show how the authors’ method improves upon current methods with respect to run-time without sacrificing accuracy with respect to estimation of the causal effect.

Weaknesses

I have no major complaints regarding this paper.

---

> ### Author Rebuttal · Authors · 2025-07-31
>
> We thank you for your thoughtful and constructive feedback. We are especially grateful for the recognition of the clarity of our writing, the novelty of our theoretical contribution, and the empirical improvements in efficiency without sacrificing accuracy. We hope the following responses properly address your concerns.
>
>
> >**Q1.** In your definition for a maximal ancestral graph, is that the same as an acyclic directed mixed graph (ADMG)? Further, what is an “almost directed cycle” in lines 90-91?
>
> **A1.** Thank you for the question.
>
> To clarify: **maximal ancestral graphs (MAGs)** are a **subclass** of **acyclic directed mixed graphs (ADMGs)**.  MAGs are used to represent the structure of data-generating processes in the presence of latent variables [Richardson, 2003; Spirtes et al., 1997] (See the first sentence of the third paragraph of [Richardson, 2003]), and they satisfy additional constraints that ADMGs do not. For example, in MAGs, **every missing edge corresponds to a conditional independence relation**, making them maximal in that sense.
>
> Regarding the term ``almost directed cycle'', it refers to a scenario where a node $X$ is both a spouse and an ancestor of another node $Y$. Specifically, $X$ is considered a spouse of $Y$ if there exists a bi-directed edge $X \leftrightarrow Y$, and an ancestor if there is a directed path from $X$ to $Y$. In an ancestral graph, the presence of a bi-directed edge $X \leftrightarrow Y$ implies that neither $X$ is an ancestor of $Y$ nor $Y$ of $X$. Therefore, allowing both a bi-directed edge and a directed path between two nodes would violate the ancestral property and create an "almost directed cycle," which is inconsistent with the semantics of MAGs [Richardson et al., 2002].
>
> **Reference**
> * Richardson T. Markov properties for acyclic directed mixed graphs. Scandinavian Journal of Statistics, 2003.
> * Spirtes P, Richardson T, Meek C. The dimensionality of mixed ancestral graphs. Technical Report CMU-PHIL-83, 1997.
> * Richardson T, Spirtes P. Ancestral graph Markov models. The Annals of Statistics, 2002.
>
>
> >**Q2.** Theorems 2,3, and 4 are nearly exactly the same as the inference rules and Theorems 2 and 3 presented in Entner et al. (2013). Is there a specific reason why the authors did not cite Entner et al. (2013) when presenting these three theorems? It seems to me that the proofs of Theorems 2, 3, and 4 should follow in a fairly straightforward manner from the proofs in Entner et al. (2013) once Theorem 1 is established. I believe that it is important to cite Entner et al. (2013) when presenting these three theorems as they are, at the very least, heavily inspired by the work of EHS.
>
> **A2.** Thank you for pointing this out.
>
> Theorems 2, 3, and 4 are indeed heavily inspired by the inderence rules proposed in Entner et al. (2013). In the revised version, we have added appropriate citations to Entner et al. (2013) when presenting these theorems.
>
> We would also like to highlight a key distinction: **our formulation is fully local**, relying only on the **Markov Blankets of the treatment and outcome variables**, rather than performing an exhaustive search over all variable combinations, as in the original inference rules. Importantly, our local approach is not merely a heuristic: it is **theoretically guaranteed to be both sound and complete** under the stated assumptions. This **localization** significantly improves computational efficiency and enhances the method's applicability in **high-dimensional settings**. We believe this constitutes a **novel and meaningful extension** beyond the original work.
>
>
>
>
> >**Q3.** Is faithfulness implicit in the theorems of this paper? Line 257 is the first mention of the faithfulness assumption in the main paper. If the authors are assuming faithfulness throughout, then they should define faithfulness and state the assumption somewhere in Section 2.
>
> **A3.** Yes, the **faithfulness assumption** is implicit in our theoretical results. Following your suggestion, we have moved its **definition and formal statement** from Appendix B to **Section 2** of the main text for greater clarity.
>
> We would also like to emphasize that **faithfulness is a standard assumption in causal discovery methods**. In fact, many classical algorithms—such as **PC**, **FCI**, and **RFCI**—**fundamentally rely on faithfulness** [Spirtes and Zhang, 2016]. Moreover, **existing covariate selection methods**, including **EHS** [Entner et al., 2013], **CEELS** [Cheng et al., 2022], and **LDP** [Maasch et al., 2024], also make use of this assumption to ensure the correctness of their independence-based inference procedures.
>
>
> Further, earlier work shows that **reducing unnecessary conditional independence tests** can help mitigate the impact of **statistically weak violations of faithfulness** [Isozaki T, 2014]. While **faithfulness remains a theoretical requirement**, our method’s **localized design naturally reduces the number and complexity of conditional independence tests**, which empirically improves robustness to potential violations of this assumption.
>
> **Reference**
> * Peter Spirtes and Kun Zhang. Causal discovery and inference: concepts and recent methodological advances. In Applied informatics, 2016.
> * Isozaki T. A robust causal discovery algorithm against faithfulness violation. Information and Media Technologies, 2014.

---

> > ### Comment · Reviewer_swW2 · 2025-08-04
> > **Response to Author Rebuttal**
> >
> > Thank you to the authors for their detailed and thoughtful response. I will respond to each of the authors' points below.
> >
> > **A1.** Thank you for clarifying the definition of almost directed cycles. I would recommend the authors to define this term either in the main text, appendix, or as a footnote. For these kinds of definitions, it may be prudent to err on the side of caution and not assume that the reader knows this term.
> >
> > **A2.** I agree with the authors' statement that their theorem is a novel and meaningful extension beyond the original work. I appreciate that the authors have added citations to Entner (2013); my concerns regarding this point have been addressed.
> >
> > **A3.** I completely agree that faithfulness is a standard assumption in causal discovery methods and covariate adjustment search methods. I am not challenging the use of faithfulness; rather, I was just hoping to clarify that the authors' method uses it. Thus, it is appropriate to state the assumption earlier in the paper, which the authors have done. My concerns regarding this point have been addressed.
> >
> > Given the authors' satisfactory responses, I will maintain my positive assessment of this paper.

---

> > > ### Author Response · Authors · 2025-08-05
> > > **Thanks**
> > >
> > > Thank you for the positive and thoughtful feedback.
> > >
> > > For A1, we appreciate your recommendation to clarify the term *almost directed cycle*. Following your advice, we have revised the manuscript to include a formal definition and a clearer explanation of its implications, ensuring that readers can easily grasp the concept.
> > >
> > > For A2, we're glad that the inclusion of the citation to Entner (2013) has addressed your concern.
> > >
> > > For A3, we have moved the statement of the **faithfulness assumption** to an earlier section of the paper, as suggested.
> > >
> > > We appreciate your recognition of these revisions. Thank you again for your constructive comments and continued support.

---

### Official Review · Reviewer_qAne · 2025-06-20

**Clarity:** 3
**Significance:** 2
**Originality:** 3
**Rating:** 5
**Confidence:** 3

**Summary:**

* The paper seems to be placed the intersection of causal discovery and causal inference -- though more focused on causal identification. In particular, the authors focus on inferring if X has a potential causal effect on Y. To this goal, they learn an appropriate set of covariates to adjust to identify and estimate causal effect; what makes this work unique is that they do it in presence of latent features.
* Relies on analyzing the testable conditional independence and dependence relationships

**Questions:**

* Can authors provide sensitivity analysis to violation of correct specification of MAG/DAG?
* In Algorithm 1, what if R1 or R2 is satisfied for multiple different combinations of S and Z? What if they result in different estimates?
* Typically, MB(Y) is always much larger than MB(X). In that case, their algorithm thats exponentially more time to run. How are authors claiming scalability?

**Ethical Concerns:**

["NO or VERY MINOR ethics concerns only"]

**Final Justification:**

The response by authors is adequate to improve my assessment of the paper

**Limitations:**

The authors discuss limitations in appendix but I think it is worth moving to maintext. Further, it will be useful to discuss the reliance on assumptions and what happens when they are violated.

**Quality:**

3

**Strengths And Weaknesses:**

Strength:
* The paper targets a relevant theoretical question -- given the causal dependencies, can we identify the causal effect and if yes, what are the set of covariates one needs to adjust for.
* The paper is very well written, with nice remarks, discussion and mix of formal and informal language

Weakness:
* The paper is notationally heavy. I am not sure there is much the authors can do.
* The work realies on stronger assumption about ordering of covariates and knowledge of MAG/DAG. I am not sure if it is realistic.

---

> ### Author Rebuttal · Authors · 2025-07-31
>
> We sincerely thank you for the thoughtful comments and for recognizing that our paper is well written, with clear remarks and discussion. Below, we address each point raised:
>
> >**W1.** The paper is notationally heavy. I am not sure there is much the authors can do.
>
> **A1.** We acknowledge that the notation may be dense at times. This is largely due to the technical nature of the identification problem under latent confounding. To assist readers, we have included a notation table (Table 1) in **Appendix B**. In the final version (thanks to the availability of an additional page), we will move this table to the **main text** to improve accessibility and readability.
>
>
>
> >**W2.** The work realies on stronger assumption about ordering of covariates and knowledge of MAG/DAG. I am not sure if it is realistic.
>
> **A2.** We would like to clarify that our method **does not assume access to the true MAG or DAG**. Instead, it is a **local, data-driven** approach that operates without relying on global structural knowledge. All experimental results are based solely on observed data, without any access to the underlying MAG or DAG.
>
> Regarding the **pretreatment assumption**, it realistically reflects how data are collected in many fields such as economics and epidemiology  [Hill, 2011, Imbens and Rubin, 2015, Wager and Athey,2018], where all covariates in $\mathbf{O}$ are measured before treatment and outcome. This assumption is particularly common in observational studies [Cheng et al., 2022, Entner et al., 2013, De Luna et al., 2011, Vander Weele and Shpitser, 2011, Wu et al., 2022]. For example:
> * In **medical studies**, covariates such as age, gender, and past medical history are collected before administering a treatment and the outcome;
> * In **economic policy evaluation**, covariates like income, education level, and demographic data are collected before a policy is implemented or the downstream outcome of interest.
>
> We adopt this assumption to focus on **covariate selection without requiring causal sufficiency**, which is a strong and often unrealistic assumption in latent variable settings.
>
> Moreover, LSAS outperforms other methods even when the pretreatment assumption may not be satisfied, as demonstrated by results on benchmark graphs (Figures 4 and 11).
>
>
> >**Q3.** Can authors provide sensitivity analysis to violation of correct specification of MAG/DAG?
>
> **A3.** We would like to clarify that our method **does not assume access to the true MAG or DAG**. LSAS is a **fully local and data-driven procedure** that identifies valid adjustment sets without requiring knowledge of the global causal structure. As such, there is **no dependency on the correct specification of the full MAG/DAG**, and hence no direct source of sensitivity to its misspecification.
>
> >**Q4.** In Algorithm 1, what if R1 or R2 is satisfied for multiple different combinations of S and Z? What if they result in different estimates?
>
> **A4.** Thank you for this valuable question.
>
> When multiple $(S, \mathbf{Z})$ pairs satisfy **condition $\mathcal{R}1$**, we follow the strategy of [Cheng et al., 2022] and **average the resulting causal effect estimates** across these valid adjustment sets. This approach helps improve stability and reduce variance.
>
> In contrast, if any $(S, \mathbf{Z})$ pair satisfies **condition $\mathcal{R}2$**, this implies that **there is no causal effect from $X$ to $Y$**. In such cases, LSAS returns a null estimate, consistent with the interpretation of R2 as a test for the absence of a causal effect.
>
> **Reference**
> * Debo Cheng, Jiuyong Li, Lin Liu, Jiji Zhang, Jixue Liu, and Thuc Duy Le. Local search for efficient causal effect estimation. IEEE Transactions on Knowledge and Data Engineering, 2022.
>
>
> >**Q5.** Typically, MB(Y) is always much larger than MB(X). In that case, their algorithm thats exponentially more time to run. **How are authors claiming scalability?**
>
> **A5.** We would like to clarify that **MB(Y) is not necessarily much larger than MB(X)** in general. The relative sizes of Markov Blankets depend heavily on the structure of the underlying graph.
>
> To **demonstrate the empirical scalability** of LSAS, we conducted additional experiments on denser random graphs, reporting both runtime and average conditioning set sizes. Specifically, we conducted additional experiments on **random graphs with increasing average degrees (3, 5, 7, and 9)**, under the same settings as in the main paper (40 nodes, sample size = 5K). These denser graphs result in larger MBs, which present a greater computational challenge.
>
> As shown in the table below, we find the follwing observations:
>
> 1. **LSAS consistently outperforms all baseline methods in terms of RE and nTests**, even as graph density increases. One exception is the (40, 9) network, where the local method **LDP** has lower nTest. While performance degrades with increasing density—as expected for all methods—LSAS maintains **notable advantages**, particularly in RE accuracy.
>
> 2. The **runtime of LSAS** is slightly higher than that of LDP in some denser settings; however, **its RE is substantially lower**, reflecting its completeness in identifying VAS. This is because we search for all adjustment sets within the Markov Blanket of $Y$. Limiting the number of valid adjustment sets found before stopping would help speed up the running time.
>
> 3. As graph density increases, the **Markov Blanket of each node grows**, leading to increased computational complexity for LSAS. This is reflected in the observed runtimes and conditioning set sizes. Importantly, this trend affects **all methods**, not just LSAS.
>
> In summary, while LSAS shares the worst-case exponential complexity inherent to constraint-based methods, its **localization strategy yields strong empirical efficiency**, especially on sparse-to-moderate graphs. We acknowledge that performance degrades with graph density, but LSAS continues to **offer better RE accuracy and competitive efficiency** compared to all baselines, even in denser settings.
>
>
> |Network|Algorithm|RE|nTest|Time (Seconds)|Conditioning Set Sizes (Mean ± std)|
> |-|-|-|-|-|-|
> |(40, 3)|LSAS|**0.123**|**174.63**|**0.014**|(2.089, 0.557)|
> ||CEELS|0.206|1738.93|0.093|(2.327, 0.819)|
> ||LDP|0.413|290.61|**0.014**|**(1.337, 0.726)**|
> ||EHS|0.362|547675|200|(5.284, 0.324)|
> |(40, 5)|LSAS|**0.275**|**403.74**|0.066|(2.510, 0.693)|
> ||CEELS|0.320|2459.63|0.312|(2.538, 0.825)|
> ||LDP|0.496|484.94|**0.059**|**(2.067, 0.752)**|
> ||EHS|0.627|412350|200|(5.217, 0.239)|
> |(40, 7)|LSAS|**0.376**|**582.62**|0.095|(2.807, 0.548)|
> ||CEELS|0.475|2548.74|0.318|(2.663, 0.705)|
> ||LDP|0.844|594.81|**0.072**|**(2.123, 0.487)**|
> ||EHS|0.827|377688|200|(5.192, 0.185)|
> |(40, 9)|LSAS|**0.407**|791.58|0.130|(3.021, 0.496)|
> ||CEELS|0.558|2594.96|0.322|(2.753, 0.527)|
> ||LDP|0.962|**618.43**|**0.104**|**(2.230, 0.421)**|
> ||EHS|0.812|361556|200|(5.180, 0.130)|
>
>
>
> > **Q6-Limitations.** The authors discuss limitations in appendix but I think it is worth moving to maintext. Further, it will be useful to discuss the reliance on assumptions and what happens when they are violated.
>
> **A6.** Thank you for the suggestion. Due to page constraints, we originally placed the discussion of limitations in the appendix, but we will move it to the main text in the final version as recommended.
>
> Regarding assumptions, our method relies on the pretreatment assumption, which, as discussed in Response A2, is widely adopted and considered realistic in many observational studies across fields such as medicine and economics.
>
> When the **pretreatment assumption is violated**, some observed covariates may be **descendants of the treatment**, potentially lying on causal paths from $X$ to $Y$. In such cases, even if $(S, \mathbf{Z})$ satisfies condition $R1$, adjusting for variables in $\mathbf{Z}$ may introduce post-treatment bias and invalidate the causal estimate. For example, in the graph:
> $$
> E \rightarrow F \rightarrow X \rightarrow Y, \quad F \leftarrow C \leftrightarrow D \rightarrow Y, \quad F \rightarrow A \rightarrow B \rightarrow Y, \quad X \rightarrow B
> $$
> if we choose $S = F$ and $\mathbf{Z} = \{B, D\}$, then $(S, \mathbf{Z})$ satisfies condition $R1$, but $\mathbf{Z}$ violates the generalized adjustment criterion since $B$ is a descendant of $X$.
>
> We will incorporate this discussion into the revised version to more clearly acknowledge the method’s assumptions and limitations.

---

> > ### Comment · Reviewer_qAne · 2025-08-07
> > **Thank you**
> >
> > I am convinced that the proposed changes by authors will make this paper stronger and I am happy to improve my assessment from 4 to 5.

---

> > > ### Author Response · Authors · 2025-08-07
> > > **Thanks**
> > >
> > > Thank you for your encouraging feedback, and we are grateful for your decision to raise the score.

---

### Official Review · Reviewer_4VeR · 2025-06-28

**Clarity:** 3
**Significance:** 2
**Originality:** 3
**Rating:** 4
**Confidence:** 3

**Summary:**

This paper focuses on causal effect estimation with latent variables. Existing methods often rely on learning the complete global causal structure, which can be computationally inefficient and unnecessary when the interest lies in a single causal effect of $X$ on $Y$. To overcome this limitation, the authors propose a novel and fully local algorithm called Local Search Adjustment Sets (LSAS). This approach leverages testable independence and dependence relationships among observed variables, specifically focusing on the Markov Blankets of $X$ and $Y$, to identify a valid adjustment set. The efficacy of the algorithm is validated through experiments on both synthetic and real-world datasets.

**Questions:**

1. I still cannot understand why the pretreatment assumption is realistic from the example "every variable within the set $O$ is measured prior to the implementation of the treatment and before the outcome is observed". Could you please provide more specific explanations?

2. While the paper mentions experimental effectiveness even when the pretreatment assumption is violated, could the authors elaborate on the theoretical implications or provide a more detailed empirical analysis of LSAS's performance in such scenarios?

3. The first step of LSAS is learning the Markov Blankets. What specific MB discovery algorithms are used? What are the computational complexities and known practical limitations of these MB discovery methods that might impact the overall efficiency of LSAS?

**Ethical Concerns:**

["NO or VERY MINOR ethics concerns only"]

**Final Justification:**

I'm not quite familiar with the scope of this paper and I found no major weaknesses. Therefore, I decide to give a positive score with a relatively low confidence.

**Limitations:**

This paper includes a section "Limitations and Future Work" in Appendix F, where they clearly state key assumptions such as the pretreatment assumption, discuss scenarios where these assumptions might be violated, and suggest future research directions to relax them, e.g., by leveraging background knowledge or data from multiple environments.

**Quality:**

3

**Strengths And Weaknesses:**

# Strengths

1. This paper proposes a novel "fully local" learning approach for covariate selection in the presence of latent variables (i.e. LSAS), distinguishing itself from methods requiring learning a global causal structure that are computationally inefficient (e.g., EHS) or incomplete (e.g., CEELS, LDP).

2. This paper proves that the proposed LSAS algorithm is both sound and complete under standard assumptions. Theorems 1, 2, 3, and 4, along with their proofs in Appendix D, establish a strong theoretical foundation for the method.

3. This paper is well-structured and clearly explains complex concepts like MAGs, Markov blankets, and the generalized adjustment criterion. The examples provided (Figures 1, 2, 3) aid in understanding the theoretical concepts and the algorithm's intuition.

4. This paper provides extensive experimental results on both synthetic and real-world datasets to demonstrate the effectiveness of the proposed LSAS.

# Weaknesses

I find no major weaknesses. Some minor questions are listed in Questions.

---

> ### Author Rebuttal · Authors · 2025-07-31
>
> We sincerely thank you for your positive evaluation and thoughtful questions. We greatly appreciate your recognition of the novelty, theoretical foundation, clarity of presentation, and comprehensive experimental results in our work. We hope that the following responses adequately address your concerns.
>
>
> >**Q1.** I still cannot understand why the pretreatment assumption is realistic from the example "every variable within the set is measured prior to the implementation of the treatment and before the outcome is observed". Could you please provide more specific explanations?
>
> **A1.** Thank you for pointing this out.
> As discussed in Remark 1 of the main paper, the **pretreatment assumption** is realistic as it reflects how data are collected in many fields such as economics and epidemiology [Hill, 2011, Imbens and Rubin, 2015, Wager and Athey,2018], where all covariates in $\mathbf{O}$ are measured before treatment and outcome. This assumption is particularly common in observational studies [Cheng et al., 2022; Entner et al., 2013; De Luna et al., 2011; Vander Weele and Shpitser, 2011; Wu et al., 2022]. For example:
> * In **medical studies**, covariates such as age, gender, and past medical history are collected before administering a treatment and the outcome;
> * In **economic policy evaluation**, covariates like income, education level, and demographic data are collected before a policy is implemented or the downstream outcome of interest.
>
>
> In such a setting, covariates are guaranteed not to be affected by the treatment or outcome, justifying the assumption. We will clarify this with concrete examples in Section 2.3 of the revised manuscript.
>
> **Reference**
>
> * Jennifer L Hill. Bayesian nonparametric modeling for causal inference. Journal of Computational and Graphical Statistics, 20(1):217–240, 2011.
> * Guido W Imbens and Donald B Rubin. Causal inference for statistics, social, and biomedical sciences: An introduction. Cambridge University Press, 2015.
> * Stefan Wager and Susan Athey. Estimation and inference of heterogeneous treatment effects using random forests. Journal of the American Statistical Association, 113(523):1228–1242, 2018.
> * Debo Cheng, Jiuyong Li, Lin Liu, Jiji Zhang, Jixue Liu, and Thuc Duy Le. Local search for efficient causal effect estimation. IEEE Transactions on Knowledge and Data Engineering, 2022.
> * Doris Entner, Patrik Hoyer, and Peter Spirtes. Data-driven covariate selection for nonparametric estimation of causal effects. In Artificial intelligence and statistics, pages 256–264. PMLR, 2013.
> * Xavier De Luna, Ingeborg Waernbaum, and Thomas S Richardson. Covariate selection for the nonparametric estimation of an average treatment effect. Biometrika, 98(4):861–875, 2011.
> * Tyler J Vander Weele and Ilya Shpitser. A new criterion for confounder selection. Biometrics, 67(4): 1406–1413, 2011.
> * Anpeng Wu, Junkun Yuan, Kun Kuang, Bo Li, Runze Wu, Qiang Zhu, Yueting Zhuang, and Fei Wu. Learning decomposed representations for treatment effect estimation. IEEE Transactions on Knowledge and Data Engineering, 35(5):4989–5001, 2022.
>
>
>
> >**Q2.** While the paper mentions experimental effectiveness even when the pretreatment assumption is violated, could the authors elaborate on the theoretical implications or provide a more detailed empirical analysis of LSAS's performance in such scenarios?
>
> **A2.** Thank you for the insightful question. When the **pretreatment assumption is violated**, some observed covariates may be **descendants of the treatment**, potentially lying on causal paths from $X$ to $Y$. In such cases, even if $(S, \mathbf{Z})$ satisfies condition $R1$, adjusting for variables in $\mathbf{Z}$ may introduce post-treatment bias and invalidate the causal estimate. For example, in the graph:
> $$
> E \rightarrow F \rightarrow X \rightarrow Y, \quad F \leftarrow C \leftrightarrow D \rightarrow Y, \quad F \rightarrow A \rightarrow B \rightarrow Y, \quad X \rightarrow B
> $$
> if we choose $S = F$ and $\mathbf{Z} = \{B, D\}$, then $(S, \mathbf{Z})$ satisfies condition $R1$, but $\mathbf{Z}$ violates the generalized adjustment criterion since $B$ is a descendant of $X$.
> A potential workaround is to identify descendants of $X$ first, then apply $R1$—but this lacks **completeness**, and may fail to recover adjustment sets even when they exist. To the best of our knowledge, **no existing method can soundly and completely identify descendants of a treatment variable locally**, especially in the presence of **latent variables** and without recovering the full graph. Addressing this remains an open and non-trivial challenge.
>
>
>
>
>
> >**Q3.** The first step of LSAS is learning the Markov Blankets. What specific MB discovery algorithms are used? What are the **computational complexities** and known **practical limitations** of these MB discovery methods that might impact the overall efficiency of LSAS?
>
> **A3.** Thank you for your question. In our implementation, we use the **TC (Total Conditioning)** algorithm [Pellet and Elisseeff, 2008b] to discover the MB; see Appendix C.2 for further details. The **computational complexity** of this method is $O(n)$, where $n$ is the number of observed variables, making it efficient for use as a first-stage procedure in LSAS.
>
> It is well known that the **accuracy of MB discovery directly affects the overall performance** of MB-based causal inference methods. In our setting, if relevant variables are **omitted** from the estimated MB, LSAS may fail to identify a VAS. To **mitigate this risk**, one can adopt a conservative approach: **relaxing the inclusion criteria** to ensure that as many true MB variables as possible are retained [Wang et al., 2014; Liu et al., 2020]. This increases the chance of including extraneous variables beyond the true MB, which in turn may **increase the computational burden** in the subsequent search phase. Nonetheless, when runtime is not a primary concern, this trade-off improves robustness and helps preserve the completeness of the adjustment set search.
>
> **Reference**
>
> * Jean-Philippe Pellet and André Elisseeff. Using Markov blankets for causal structure learning. Journal of Machine Learning Research, 9(7), 2008b.
> * Wang, C., Zhou, Y., Zhao, Q., & Geng, Z. Discovering and orienting the edges connected to a target variable in a DAG via a sequential local learning approach. Computational Statistics & Data Analysis, 2014.
> * Liu, Y., Fang, Z., He, Y., Geng, Z., & Liu, C. Local causal network learning for finding pairs of total and direct effects. Journal of Machine Learning Research, 2020.

---

> > ### Comment · Reviewer_4VeR · 2025-08-04
> >
> > I appreciate the author's rebuttal. I decide to maintain my positive score.

---

> > > ### Author Response · Authors · 2025-08-04
> > > **Thanks**
> > >
> > > Thank you very much for your positive feedback and appreciation of our work!

---

### Official Review · Reviewer_urL5 · 2025-07-01

**Clarity:** 3
**Significance:** 3
**Originality:** 3
**Rating:** 5
**Confidence:** 4

**Summary:**

This paper addresses the problem of identifying a valid adjustment set from observational data, opting to use a local, rather than global, causal discovery method. Assuming that the observed covariates O satisfy the pretreatment assumption, and additionally that Y is not a causal ancestor of X, the authors propose a local learning algorithm LSAS that identifies a valid adjustment set when possible (being both sound and complete). Their algorithm employs a two-stage procedure: first the Markov Blankets of X and Y are identified, then variables are added to the adjustment set if they satisfy the proposed rules R1 and R2. The authors then apply LSAS to both real and synthetic data, demonstrating that the VAS produced by LSAS attain the lowest relative error and lowest number of tests run when compared to some baselines.

**Questions:**

Q1. I find it unclear why the pretreatment assumption is maintained and what specific challenges prevent generalization to non-pretreatment settings. Can you provide:
- A concrete explanation of what algorithmic or theoretical barriers exist in the non-pretreatment case?
- Specific examples showing how the LSAS approach would fail without this assumption?
- A clear characterization of what types of problematic variables (beyond m-structures) LSAS aims to exclude?

Q2. I am concerned that using nTests as a proxy for computational complexity may be misleading, since conditioning set sizes could vary significantly between methods. Can you:
- Report actual wall-clock runtime comparisons alongside nTests for all baselines?
- Analyze whether LSAS systematically uses smaller/larger conditioning sets than competitors?
- Provide theoretical or empirical analysis of how conditioning set sizes scale with problem dimensions

Q3. The experimental evaluation primarily focuses on sparse graphs (average degree 3), but limited results on denser graphs (INSURANCE/ANDES) show LSAS performing similarly to baselines. Can you:
- Conduct systematic experiments on synthetic dense graphs with varying average degrees (5, 7, 10+)?
- Provide theoretical analysis of how LSAS complexity scales with graph density?
- If performance degrades on dense graphs, acknowledge this limitation and characterize the settings where LSAS provides advantages?

If the authors adequately address my concerns, in both the Strengths and Weaknesses and Questions sections, I am open to raising my score.

**Ethical Concerns:**

["NO or VERY MINOR ethics concerns only"]

**Final Justification:**

The authors thoroughly addressed the concerns I had in my review, including good clarifications about the novelty of their method, as well as adding crucial experiments that validate the significance of their approach's performance and runtime in dense settings. I recommend acceptance.

**Limitations:**

See significance subsection of Strengths and Weaknesses section.

**Paper Formatting Concerns:**

None.

**Quality:**

3

**Strengths And Weaknesses:**

Strengths:

Quality - The work seems to me to be technically sound - although I did not inspect the proofs in the appendix closely, the high-level explanations given for the main results (such as Theorem 2 and Theorem 3) seem correct. The strategy of searching for conditions that allow one to conclude that there does exist a VAS, or that a VAS cannot be determined is intuitive. The work is complete in the sense that Theorem 5 (soundness/completeness) shows that LSAS is effective in the maximal number of cases in this setting.

Originality - It is clear the LSAS represents a step-forward from prior VAS identification methods, as the first one to satisfy both completeness and soundness without necessarily learning the strong graph (albeit under the strong pre-treatment assumption). The authors make this clear in both the second half of the introduction, as well as explicit remarks in Section 2.3. The reasoning behind the approach, i.e., identifying the necessary conditions for identifiability of VAS, then checking for whenever these conditions are satisfied, makes intuitive sense.


Weaknesses:

Clarity - It is not really clear why the authors want to keep the pretreatment assumption. Its not clear what the challenge of non-pretreatment assumption is setting is, or why their current approach cannot be generalized to the non-pretreatment setting. Even under the pretreatment assumption, it is not clear to the reader what the main difficulty is that their algorithm tries to overcome. For example, without latent confounders the entire covariate set O constitutes a VAS under the pretreatment assumption, as there are no mediators or colliders. However, with pretreatment, one issue that arises are nodes that satisfy an m-structure induce a backdoor path when conditioned on - its not clear if this is really the only variables we want to exclude or if there other types of problematic variables  to exclude. Given these issues, it is difficult to ascertain exactly what insights researchers should derive about when local discovery approach of LSAS is intuitive, what are the major blockers, perhaps in some domains how it can be overcome.


Significance - Although LSAS is a local (rather than global) VAS identification method that does not relax the strong pretreatment assumption, it still retains a worst-case exponential (in dimension) runtime bound common to many global methods. This severely limits its applicability in practice, as it seems likely that LSAS would fall victim to a common practioner complaint, which is that many constraint-based discovery algorithms never converge outside small toy datasets. Although the experimental evaluation of LSAS presented in the main text seems to dispel those concerns, as LSAS seems to outperform other methods by large margins in both RE accuracy and nTests, I have strong concerns about whether these results honestly represent the capabilities of LSAS.

The authors note that the global VAS identification method EHS has already been shown to be sound and complete, but with time complexity exponential in the number of covariates. This excessive runtime motivates the local search approach of LSAS, and the authors show theoretically that LSAS achieves worst time complexity exponential in the Markov Blanket of $Y$, rather than all covariates, a definitive improvement. However, in their experiments, the authors use the number of tests ran nTests, rather than the absolute runtime of the algorithms (seconds), as an empirical measure of time complexity. This can be misleading, as the number of tests can diverge from absolute runtime, when the size of conditioning sets increases. It is not clear whether LSAS is prone to using smaller conditioning set sizes, therefore it is inappropriate to use nTest as a measure of time complexity, as it could mislead readers when a method runs fewer tests, but actually runs for a much longer time. The authors should take care to report runtime metrics alongside nTests for all baselines, to ensure fair comparison.

Further, the experimental results in the appendix present some additional worries. While experiments on synthetic data with randomly generated graphs (Figure 10) with increasing dimensionality ($d= 20,30,40,50) show that LSAS outperfoming baselines significantly, all settings use only sparse causal graphs, with a constant average degree 3 for all nodes. Given that the complexity of LSAS depends largely on its first step of Markov Blanket identification, if the MB of Y is small (which would be the case for sparse graphs), LSAS would enjoy a large improvement over baseline methods. However, this leaves the performance of LSAS in denser graphs entirely uncharacterized. This worry is further supported by the experimental results on the real-world INSURANCE and ANDES datasets in Figure 11; INSURANCE and ANDES have higher average degrees than the benchmarks commented on in the main text (3.03,3.85 vs 2.63,2.95), and here LSAS performs nearly identically to CEELS with respect to RE. Given a limited empirical analysis of LSAS's performance on dense graphs, and the few relevant experiments being relegated to the end of the appendix, this obscures the significance of LSAS to the broader community.

---

> ### Author Rebuttal · Authors · 2025-07-31
>
> We thank you for the thoughtful feedback and appreciate the positive remarks on the soundness and originality of our work. We address your concerns below.
>
> > **W1&Q1.** the pretreatment assumption...generalization to non-pretreatment settings...fail without this assumption...beyond m-structures LSAS aims to exclude.
>
> **A1.** We would like to clarify the motivation and scope of our work as follows.
> * First, regarding the pretreatment assumption, it realistically reflects how data are collected in many fields such as economics and epidemiology  [Hill, 2011, Imbens and Rubin, 2015, Wager and Athey,2018]. We adopt this assumption to focus on **covariate selection without requiring causal sufficiency**. While the full set $\mathbf{O}$ forms a VAS in the absence of latent confounders, conditioning on all covariates—especially in high-dimensional settings—can lead to poor performance [Luna et al., 2011; Abadie and Imbens, 2006]. Our goal is to identify the *local sufficient VAS* that enables unbiased and efficient causal effect estimation, **even in the presence of latent variables**.
>
> * Second, even under the pretreatment assumption, **selecting a VAS remains non-trivial** in the presence of latent confounders. Not all pretreatment covariates are harmless to condition on—for example, variables involved in m-structures or other types of collider paths (e.g., $X \leftrightarrow A \leftrightarrow B \leftrightarrow C \leftrightarrow Y$) may introduce bias by opening backdoor paths. Our method identifies which covariates to include or exclude based on their roles in the local causal structure, which is learned directly from data without requiring global graph recovery.
>
> * Third, existing methods-such as EHS, CEELS, and ours-**do not directly extend** to settings where the **pretreatment assumption is violated**, as they may select invalid adjustment sets that include descendants of the treatment, leading to biased estimates. For example, in the graph:
> $$
> E \rightarrow F \rightarrow X \rightarrow Y, \quad F \leftarrow C \leftrightarrow D \rightarrow Y, \quad F \rightarrow A \rightarrow B \rightarrow Y, \quad X \rightarrow B
> $$
> if we choose $S = F$ and $\mathbf{Z} = \{B, D\}$, then $(S, \mathbf{Z})$ satisfies condition $R1$, but $\mathbf{Z}$ violates the generalized adjustment criterion since $B$ is a descendant of $X$.
> A potential workaround is to identify descendants of $X$ first, then apply $R1$—but this lacks **completeness**, and may fail to recover adjustment sets even when they exist. To the best of our knowledge, **no existing method can soundly and completely identify descendants of a treatment variable locally**, especially in the presence of **latent variables** and without recovering the full graph. Addressing this remains an open and non-trivial challenge.
>
> * Finally, as discussed in Section 1 and Appendix A, existing global adjustment set methods are often inefficient, while existing local approaches suffer from **incompleteness**—even under the pretreatment assumption. This highlights the need for a method that is both **sound and complete**, operates **locally**, and remains valid in the presence of **latent variables**, to reliably select the VAS for specific causal queries.
>
> 1. Abadie A, Imbens G W. Large sample properties of matching estimators for average treatment effects. Econometrica, 2006.
>
>
> > **W2&Q2.** ...report runtime metrics alongside nTests... Analyze conditioning sets...theoretical or empirical analysis of how conditioning set sizes scale with problem dimensions?
>
> **A2.** Thank you for the thoughtful comments.
>
> We agree that nTests alone may not fully reflect runtime performance, especially when conditioning set sizes differ across methods. As suggested, we have now included both wall-clock runtimes and average conditioning set sizes for all benchmark networks. Due to high computational cost, EHS did not finish within 2 hours; we report its runtime and conditioning set sizes using a 200-second early stopping threshold for fair comparison.
>
> Our key observations are as follows:
> 1. EHS, as a global search method, exhibits the **highest runtime and largest conditioning sets** across all settings, as expected.
> 2. LSAS consistently outperforms baselines in runtime across most graphs and sample sizes. One exception is the WIN95PTS network at large sample sizes (15K), where the local method **LDP** is faster; however, **LSAS achieves substantially higher RE accuracy** (see Figures 4 and 11), due to LDP’s incompleteness in identifying valid adjustment sets.
> 3. Among the three local search methods, **LSAS uses slightly larger conditioning sets on average**. This is because LSAS is designed to search for **all valid adjustment sets within $MB(Y)$**, rather than stopping after identifying the first one. **Despite this overhead**, LSAS achieves **consistently lower RE** than CEELS and LDP (see Figures 4 and 11), **highlighting the advantage of its completeness in VAS identification**.
>
> In summary, these results suggest that LSAS offers a favorable trade-off between runtime and estimation accuracy.
>
> |Network|Size|Algorithm|Time(Seconds)|Conditioning Set Sizes(Mean ± std)
> |-|-|-|-|-|
> |ANDES|1K|LSAS|**0.035**|(2.585, 0.744)|
> |||CEELS|0.337|(1.521, 0.758)|
> |||LDP|0.307|(1.835, 1.654)|
> |||EHS|200|(3.044, 0.096)|
> ||5K|LSAS|**0.048**|(2.619, 0.760)|
> |||CEELS|1.466|(1.843, 0.910)|
> |||LDP|0.456|(1.614, 1.313)|
> |||EHS|200|(3.065, 0.119)|
> ||10K|LSAS|**0.059**|(2.640, 0.778)|
> |||CEELS|1.752|(2.147, 0.991)|
> |||LDP|0.477|(1.494, 1.277)|
> |||EHS|200|(3.086, 0.130)|
> ||15K|LSAS|**0.068**|(2.646, 0.776)|
> |||CEELS|1.634|(2.140, 0.970)|
> |||LDP|0.563|(1.531, 1.234)|
> |||EHS|200|(3.165, 0.157)|
> |INSURANCE|1K|LSAS|**0.007**|(2.131, 0.509)|
> |||CEELS|0.253|(1.583, 0.494)|
> |||LDP|0.028|(2.25, 0.577)|
> |||EHS|200|(6.857, 0.471)|
> ||5K|LSAS|**0.015**|(2.269, 0.631)|
> |||CEELS|0.442|(1.999, 0.580)|
> |||LDP|0.031|(2.305, 0.595)|
> |||EHS|200|(6.827, 0.446)|
> ||10K|LSAS|**0.020**|(2.325, 0.704)|
> |||CEELS|0.540|(2.229, 0.682)|
> |||LDP|0.033|(2.263, 0.602)|
> |||EHS|200|(6.844, 0.466)|
> ||15K|LSAS|**0.023**|(2.326, 0.685)|
> |||CEELS|0.687|(2.420, 0.713)|
> |||LDP|0.037|(2.315, 0.596)|
> |||EHS|200|(6.858, 0.434)|
> |MILDEW|1K|LSAS|**0.006**|(2.175, 0.409)|
> |||CEELS|0.170|(1.309, 0.554)|
> |||LDP|0.026|(1.007, 0.487)|
> |||EHS|200|(5.455, 0.436)|
> ||5K|LSAS|**0.009**|(2.157, 0.443)|
> |||CEELS|0.243|(1.691, 0.849)|
> |||LDP|0.026|(0.968, 0.420)|
> |||EHS|200|(5.434, 0.398)|
> ||10K|LSAS|**0.010**|(2.160, 0.454)|
> |||CEELS|0.343|(1.999, 0.816)|
> |||LDP|0.036|(1.067, 0.508)|
> |||EHS|200|(5.489, 0.409)|
> ||15K|LSAS|**0.011**|(2.158, 0.459)|
> |||CEELS|0.428|(2.010, 0.719)|
> |||LDP|0.031|(1.039, 0.500)|
> |||EHS|200|(5.396, 0.395)|
> |WIN95PTS|1K|LSAS|**0.011**|(2.215, 0.913)|
> |||CEELS|0.058|(1.298, 0.421)|
> |||LDP|0.028|(0.388, 0.161)|
> |||EHS|200|(3.826, 0.086)|
> ||5K|LSAS|**0.028**|(2.718, 1.095)|
> |||CEELS|0.213|(1.597, 0.43)|
> |||LDP|0.035|(0.443, 0.254)|
> |||EHS|200|(3.839, 0.161)|
> ||10K|LSAS|**0.035**|(2.834, 1.139)|
> |||CEELS|0.332|(1.673, 0.382)|
> |||LDP|0.040|(0.465, 0.30)|
> |||EHS|200|(3.758, 0.162)|
> ||15K|LSAS|0.041|(2.933, 1.121)|
> |||CEELS|0.406|(1.702, 0.375)|
> |||LDP|**0.038**|(0.485, 0.332)|
> |||EHS|200|(3.795, 0.163)|
>
> > **W3&Q3.** ...the performance of LSAS in denser graphs...how LSAS complexity scales with graph density?...characterize the settings where LSAS provides advantages?
>
> **A3.** Thank you for the valuable suggestion. We conducted additional experiments on random graphs with average degrees 3, 5, 7, and 9  (40 nodes, 5K samples), as in the main paper. Due to high computational cost, EHS did not finish within 2 hours; we report its runtime and conditioning set sizes using a 200-second early stopping threshold for fair comparison.
>
> Our key observations are as follows:
>
> 1. As shown in the table below, **LSAS consistently outperforms all baseline methods in terms of RE and nTests**, even as graph density increases. One exception is the (40, 9) network, where the local method **LDP** has lower nTest. While performance degrades with increasing density—as expected for all methods—LSAS maintains **notable advantages**, particularly in RE accuracy.
> 2. The **runtime of LSAS** is slightly higher than that of LDP in some denser settings; however, **its RE is substantially lower**, reflecting its completeness in identifying VAS. This is because we search for all adjustment sets within the Markov Blanket of $Y$. Limiting the number of valid adjustment sets found before stopping would help speed up the running time.
> 3. As graph density increases, the **Markov Blanket of each node grows**, leading to increased computational complexity for LSAS. This is reflected in the observed runtimes and conditioning set sizes. Importantly, this trend affects **all methods**, not just LSAS.
>
> In summary, while LSAS shares the worst-case exponential complexity of constraint-based methods, its **localization strategy yields strong empirical efficiency**, especially on sparse-to-moderate graphs. While performance degrades with increasing density—as with all methods—LSAS consistently delivers better RE accuracy and competitive efficiency across all settings.
>
> |Network|Algorithm|RE|nTest|Time(Seconds)|Conditioning Set Sizes(Mean ± std)|
> |-|-|-|-|-|-|
> |(40, 3)|LSAS|**0.123**|**174.63**|**0.014**|(2.089, 0.557)|
> ||CEELS|0.206|1738.93|0.093|(2.327, 0.819)|
> ||LDP|0.413|290.61|**0.014**|**(1.337, 0.726)**|
> ||EHS|0.362|547675|200|(5.284, 0.324)|
> |(40, 5)|LSAS|**0.275**|**403.74**|0.066|(2.510, 0.693)|
> ||CEELS|0.320|2459.63|0.312|(2.538, 0.825)|
> ||LDP|0.496|484.94|**0.059**|**(2.067, 0.752)**|
> ||EHS|0.627|412350|200|(5.217, 0.239)|
> |(40, 7)|LSAS|**0.376**|**582.62**|0.095|(2.807, 0.548)|
> ||CEELS|0.475|2548.74|0.318|(2.663, 0.705)|
> ||LDP|0.844|594.81|**0.072**|**(2.123, 0.487)**|
> ||EHS|0.827|377688|200|(5.192, 0.185)|
> |(40, 9)|LSAS|**0.407**|791.58|0.130|(3.021, 0.496)|
> ||CEELS|0.558|2594.96|0.322|(2.753, 0.527)|
> ||LDP|0.962|**618.43**|**0.104**|**(2.230, 0.421)**|
> ||EHS|0.812|361556|200|(5.180, 0.130)|

---

> ### Comment · Reviewer_urL5 · 2025-08-04
>
> I thank the authors for their thoughtful response and for addressing the points I raised. I will increase my score to weak accept (4), and am inclined to raise it further if the authors agree to incorporate the additional experimental results present in answers A2 and A3, as well as the discussion in A1, into the camera ready version of this paper. Such results are crucial to contextualize the strong performance of LSAS, relative to baselines, across different contexts.

---

> > ### Author Response · Authors · 2025-08-05
> > **Thanks**
> >
> > We sincerely thank you for your positive feedback and for considering an increased score. We appreciate your recognition of the significance of the additional experimental results (A2 and A3) and the discussion in A1.
> >
> > We **confirm** that we will incorporate the experimental results and the corresponding discussion into the camera-ready version. Specifically:
> >
> > * **From A1:**
> >
> >   * We will include a discussion of the **pretreatment assumption** and its role in observational studies in **Section 2.3 (Problem Setup)**.
> >   * We will clarify why selecting a valid adjustment set remains non-trivial **even under the pretreatment assumption** when latent confounders are present, also in **Section 2.3**.
> >   * The example showing that existing methods (EHS, CEELS, and ours) do not directly extend to settings where the **pretreatment assumption is violated** will be included in **Section``Limitations and Future Work''**. **Thanks to the availability of an additional page, we will move this section into the main text in the final version.**
> >
> > * **From A2 and A3:**
> >
> >   * The additional **runtime and conditioning set size comparisons** on benchmark graphs will be included in **Section 4.1 (Synthetic Data-Benchmark Networks)**, along with an explanation of conditioning set size.
> >   * The **new experiments on graphs with increasing density** and the accompanying analysis will be integrated into **Section 4.1 (Synthetic Data)**.
> >   * A discussion of **limitations** (LSAS shares the worst-case exponential complexity of constraint-based methods) and **strengths** (its localization strategy yields strong empirical efficiency) under varying graph densities will be added to **Section 5 and Section``Limitations and Future Work''**.
> >
> > We fully agree that these additions will enhance the clarity and impact of our work. Thank you again for your constructive and encouraging review.

---

### Decision · Program_Chairs · 2025-09-17

**Decision:**

Accept (poster)

**Comment:**

The paper proposes a local sound and complete algorithm for identifying valid adjustment sets under latent confounding. Its main strengths are rigorous theory, clear exposition, and improved efficiency over global methods together with extensive experiments. Reviewers’ concerns on assumptions, runtime, and dense graphs were addressed with additional results and clarifications in the discussion phase so that eventually all reviewers recommend acceptance and I fully agree with their assessment.